# Click Chemistry-Based Hydrogels for Tissue Engineering

**DOI:** 10.3390/gels11090724

**Published:** 2025-09-11

**Authors:** Soheil Sojdeh, Amirhosein Panjipour, Amal Yaghmour, Zohreh Arabpour, Ali R. Djalilian

**Affiliations:** Department of Ophthalmology and Visual Science, University of Illinois, Chicago, IL 60612, USA; sojdesoheil@gmail.com (S.S.);

**Keywords:** click chemistry, crosslinked hydrogel, tissue engineering

## Abstract

Click chemistry has become a powerful and flexible approach for designing hydrogels used in tissue engineering thanks to its high specificity, fast reaction rates, and compatibility with biological systems. In this review, we introduce the core principles of click chemistry, including efficiency, orthogonality, and modularity, and highlight the main types of reactions commonly used in hydrogel formation, such as azide-alkyne c-cloadditions, thiol-ene/yne reactions, Diels–Alder cycloadditions, and tetrazine–norbornene couplings. These chemistries allow researchers to create covalently crosslinked hydrogels that are injectable, responsive to environmental stimuli, biodegradable, or multifunctional. We also explore strategies to enhance bioactivity, such as incorporating peptides, growth factors, or extracellular matrix components, and enabling precise spatial and temporal control over biological cues. Click-based hydrogels have shown promise across a wide range of tissue engineering applications, from cartilage and skin repair to neural regeneration, corneal healing, and cardiovascular scaffolds, as well as in 3D bioprinting technologies. Despite the many advantages of click chemistry such as mild reaction conditions and customizable material properties, some challenges remain, including concerns around copper toxicity, the cost of specialized reagents, and scalability. Finally, we discuss the status of clinical translation, regulatory considerations, and future directions, including integration with advanced bio fabrication methods, the design of dual-click systems, and the emerging role of in vivo click chemistry in creating next-generation biomaterials.

## 1. Introduction

Hydrogels are polymeric materials with three-dimensional network structures, whose unique combination of high water content, tunable mechanical properties, and biocompatibility has led to a wide range of applications in various fields, especially in biomedicine [1]. Hydrogels closely mimic the mechanical and physicochemical properties of human soft tissues, making them highly suitable for applications such as drug delivery, wound healing, and tissue repair [2]. Beyond the biomedical field, hydrogels have attracted attention in a variety of fields, generating innovations in polymer chemistry and crosslinking strategies [3]. For example, agricultural applications have led to the development of stimuli-responsive hydrogels for the controlled release of agricultural chemicals, while research in food preservation and biotechnology has emphasized the use of sustainable biopolymer-based systems. These advances have introduced design principles such as responsiveness to environmental cues and eco-friendly crosslinking that are now being adapted for use in biomedicine. Click chemistry has played a key role in translating these concepts into safer and more precise biomedical hydrogels, allowing for controllable degradation, biorthogonal reactivity, and tunable functionality suitable for clinical applications [4].

Hydrogels can be physically or chemically crosslinked. Physically crosslinked hydrogels are stabilized by noncovalent interactions such as ionic bonds, hydrogen bonds, or polymer entanglements, whereas chemically crosslinked hydrogels rely on covalent bonds and provide greater mechanical stability and durability [5]. Common chemical crosslinking methods include free radical polymerization, enzyme-induced crosslinking, and “click” chemistry. Compared with physical crosslinking, chemically crosslinked hydrogels exhibit superior mechanical properties and greater stability, expanding their potential applications. While conventional crosslinking methods have been extensively investigated over the past century, opportunities for innovative innovations remain limited. Many traditional chemical strategies have reached a plateau due to challenges such as cytotoxicity of residual crosslinkers, poor tunability of mechanical and degradation profiles, and lack of specificity in biological interactions [6].

In recent years, click chemistry has emerged as a highly versatile and powerful strategy for hydrogel synthesis. Click reactions are rapid, selective, high-yielding, and often produce minimal or benign side products under mild conditions, making them well-suited for biomedical applications. These features have enabled the creation of a wide range of functional materials with applications in medicine, agriculture, and biotechnology [7]. In hydrogel systems, click chemistry offers precise control over network architecture, mechanical properties, and biological function, while supporting biorthogonal reactions compatible with physiological environments. Early examples include the pioneering work of Hubbel and colleagues who used Michael addition click reactions to produce bioactive hydrogels for tissue repair. Since then, numerous click-based strategies have been developed to fabricate hydrogels with tunable stiffness, degradability, and bio functionality for diverse biomedical applications [8].

While the application of click chemistry for fabricating hydrogels has been extensively documented in previous reviews, with their focus largely on fundamental reactions, basic material properties, and established models, this review builds upon that foundation to offer a distinct and timely contribution. We move beyond these established concepts to critically explore the cutting-edge potential of click-crosslinked hydrogels, with a distinct focus on emerging applications that point toward the future of the field. This includes a dedicated exploration of their transformative role in precision medicine, spatiotemporally controlled drug delivery systems, and the bio fabrication of hierarchically complex tissues. To contextualize their versatility, we first introduce common click reactions and highlight their parallel advancements in fields like nucleic acid chemistry, a connection often overlooked in tissue engineering-focused articles. Furthermore, we synthesize these insights to provide a forward-looking roadmap, offering practical design considerations and clear future perspectives aimed at empowering researchers to develop the next-generation of clinically impactful and translational hydrogel-based therapies.

## 2. Fundamentals of Click Chemistry in Hydrogel Design

Tissue regeneration is facilitated by intelligent biomaterials that include biological components. Biomaterials spatially localize and control the release of growth factors while promoting cell migration and proliferation. Tissue regeneration requires the development and building of sophisticated biomaterial platforms. Hydrogels are among the most significant educational biomaterials because of their three-dimensional (3D) water-saturated polymeric networks [9,10].

Crosslinking hydrogels may be performed chemically or physically [11]. Crosslinked hydrogels are physically connected by noncovalent interactions such molecular entanglements, hydrogen bonds, and ionic bonds [12]. Compared to physical hydrogels, chemically crosslinked hydrogels have stronger covalent connections. Enzyme-induced crosslinking, “click” chemistry, and free-radical polymerization are the main mechanisms [13]. Hydrogel networks that are chemically crosslinked have better mechanical qualities, increased stability, and a wider variety of uses [14]. Recently, “click” chemistry was found and is developing as illustrated in Figure 1. It is the best option for hydrogel crosslinking because of its outstanding research value, quick reaction time, and specificity.

As illustrated in Figure 1, the principal mechanistic pathways for fabricating hydrogels via click chemistry, a suite of bioorthogonal reactions prized for their high efficiency, specificity, and mild reaction conditions. The illustration highlights key reactions, including the Strain-Promoted Azide-Alkyne Cycloaddition (SPAAC), which facilitates rapid, cytocompatible crosslinking without the need for cytotoxic copper catalysts, a significant advantage over conventional copper-catalyzed azide-alkyne cycloaddition (CuAAC). Furthermore, it details thiol-based click reactions, such as thiol-ene, which are initiated by UV/Vis light under physiological conditions, offering precise spatiotemporal control over gelation, a feature largely unattainable with traditional thermal initiators [15,16]. The inclusion of the Diels–Alder reaction further exemplifies a thermally reversible cycloaddition that enables the formation of hydrogels with tunable and often self-healing viscoelastic properties. Systematically, these click mechanisms offer distinct performance advantages over conventional crosslinking methods (e.g., free-radical polymerization, glutaraldehyde treatment). They typically proceed with high fidelity and quantitative yields, resulting in hydrogels with more predictable and uniform network structures, which directly translates to superior control over mechanical properties and degradation profiles. Crucially, their bioorthogonal nature often eliminates the formation of undesirable byproducts, mitigating the cytotoxicity and inflammatory responses commonly associated with conventional chemical crosslinkers [17,18]. This combination of precise spatial-temporal control, enhanced biocompatibility, and excellent reproducibility makes click chemistry not merely an alternative method, but a transformative approach for generating well-defined, functional polymer networks. These networks are particularly powerful for advanced biomaterial design, enabling applications in cell delivery, 3D bioprinting, and dynamic tissue engineering scaffolds that demand a high degree of precision and compatibility [19].

As a result, it is one of the most widely used methods for producing hydrogel for regenerative medical applications such as wound healing, bone regeneration, spinal cord regeneration, cartilage regeneration, and corneal regeneration, hydrogels have shown promise [20,21]. Hydrogels with different physical, chemical, and biological characteristics may be crosslinked and chemically functionalized in a variety of synthetic ways according to this chemical approach. In the 1990s, Hubbell and colleagues developed Michael addition hydrogels for tissue regeneration [22]. Biomedical hydrogels have recently been synthesized by many researchers using novel click reactions [8].

### Definition and Key Principles of Click Chemistry

In order to develop biohydrogels for tissue healing, Hubbell and associates were the first to use click chemistry for hydrogel production [23]. Sharpless and colleagues coined the term “click chemistry” in 2001 [24]. The concept was quickly adopted, representing a major breakthrough in synthetic chemistry and inspiring scientists in almost every area of chemistry [25]. Since its beginning, “click” chemistry has helped develop several useful chemicals and innovative materials with uses in agriculture, health, and other fields [26,27,28,29].

“Click” chemistry hydrogels are becoming more and more popular in the hydrogel space because of their improved selectivity and physiologically induced biorthogonal reactions [17]. A spontaneous, fast, highly selective, and high-yield chemical reaction between two molecules under moderate conditions is referred to as “click” chemistry [30]. For instance, the production of hydrogel results in either no byproducts or only water [31]. One useful bio-orthogonal method for creating chemically crosslinked hydrogels is click chemistry. The various click reactions used to crosslink polymer networks are summarized in Table 1.

## 3. Major Click Reactions Used in Hydrogels

Over the last decade, advancements in hydrogel production techniques have enabled researchers to alter or synthesize innovative polymers with multifunctional characteristics for applications in tissue engineering [44,45,46,47]. According to Sharpless’ research group, click chemistry reactions should be modular, diverse, high-yielding, stable under physiological settings, stereospecific, and non-toxic, using easily accessible materials and reagents under simple reaction conditions. The reaction must provide products instantaneously without the need for chromatography or toxic solvents [48]. Copper-based click chemistry was first used for hydrogel synthesis; however, its application in tissue engineering and regenerative medicine diminished due to the toxicity of copper ions and the generation of reactive oxygen species [16].

Consequently, researchers choose copper-free click chemistry for the production of hydrogels in tissue engineering and regenerative medicine [49]. Copper-free click chemistry serves several tasks without the need of detrimental catalysts or toxins post-gel formation [50]. Copper-free click chemistry methodologies including SPAAC click hydrogels, Diels–Alder click hydrogels, thiol-ene, oxime, and thiol-yne reactions [51,52,53]. Alternative approaches for synthesizing click chemistry hydrogels include pseudo click reactions, which provide high yields in moderate and highly reactive conditions [17]. These hydrogel materials, based on click chemistry, have significant potential for 3D bioprinting of tissue and organ structures [54,55,56]. For example, Cheng et al. developed an injectable lubricative hydrogel based on a hydration-lubrication mechanism and dynamic disulfide bonds via a pseudo-click reaction for tendon repair [57]. Similarly, Ye et al. functionalized a linear DNA strand with a responsive luminescent active molecule through disulfide bonds for wound-healing applications [58]. Although scaffold-free methods are being explored for tissue regeneration, injectable in situ hydrogels, with or without live cells, growth factors, biomolecules, nanoparticles, and microspheres, demonstrate potential for tissue engineering due to enhanced mechanical properties and biocompatibility (Figure 2).

### 3.1. Azide–Alkyne Cycloaddition (Cu/SPAAC)

A terminal alkyne and an aliphatic azide, when combined with a copper catalyst, produce a 1,4-disubstituted 1,2,3-triazole. Cycloaddition processes, although exothermic, exhibit a significant activation barrier for the reagent compounds, leading to diminished reaction rates even at elevated temperatures. Copper (I) facilitates gelation at physiological temperature, with durations varying from seconds to hours based on catalyst or monomer concentration and the structural characteristics of the polymer components [59]. CuAAC crosslinks exhibit functional stability under standard physiological settings but deteriorate in acidic environments. The degradation of CuAAC-crosslinked scaffolds is contingent upon the inherent degradability of the polymers rather than the stability of the crosslinks. For example, the functionalization of gelatin with alkyne groups and the combination of precursor polymers with a small molecule PEG-diazide in the presence of copper yielded hydrogels that exhibited hydrolytic stability in PBS buffer at 37 °C for 7 weeks, while CuAAC-based hydrogels remained completely intact after 2 weeks in vivo [60]. Copper ions exhibit cytotoxicity at quantities beyond micromolar levels, posing challenges in scenarios requiring expedited gelation, often facilitated by elevated copper concentrations [61]. Consequently, while CuAAC offers benefits in stability, tunability, bio-orthogonality, and a reasonably quick reaction rate, its cytotoxicity poses challenges. CuAAC-based click chemistry generates many hydrogels within a brief gelation timeframe; nevertheless, the toxicity of the copper catalyst renders them inappropriate for tissue engineering and regenerative medicine. Nevertheless, the effective modification of catalysts, the use of chelating agents, and the creation of alternative copper catalysts may provide less hazardous CuAAC-based hydrogels for tissue engineering [17,62]. The use of CuAAC scaffolds is limited to synthetic polymers, since they cannot be eliminated from the body without chemical modifications to the polymer backbones [63]. The generation of cytotoxic copper ions from the catalyst during the process restricts its use in tissue engineering and regenerative medicine [64].

Numerous investigations used water-soluble chelating agents or ligands such as bipyridine, bis(L-histidine), and bis [tert-butyl triazoyl) methyl] to address this challenge. Guo et al. synthesized mussel-inspired citrate-based bio-adhesive hydrogels using 4-(2-hydroxyethyl)-1-piperazineethanesulfonic acid as a copper chelating agent in CuAAC procedures. These hydrogels exhibit excellent bioadhesive properties and robust antibacterial efficacy, making them appropriate for invasive applications. As the concentration of periodate increased, the length of gelation decreased somewhat, presumably owing to crosslinking of catechol groups rather than click crosslinking [64,65,66]. In another study, Bertozzi’s group identified copper-free “click” chemistry that use cyclooctane to react with azide groups, producing aromatic triazoles at ambient temperature without a catalyst. The structure of cyclooctane dictates reaction kinetics in the absence of a catalyst. In contrast to the CuAAC reaction, this reaction maintains a very high reaction rate while exhibiting significant chemical selectivity and biocompatibility. This indicates that the reaction has significant promise and application in biomedicine [67]. Researchers devised click reactions that do not need copper catalysts to mitigate these limitations. This resulted in strain-promoted azide-alkyne cycloaddition click reactions. “Strain-promoted” refers to the ring strain that accelerates azide-cyclooctyne reactions relative to other methods. The Strain-Promoted Azide-Alkyne Cycloaddition (SPAAC) process employs a highly strained cyclooctyne precursor to reduce the activation energy of the alkyne-azide reaction, facilitating triazole crosslinking in the absence of a copper catalyst [49]. Han et al. formulated an SPAAC-crosslinked hyaluronic acid-based hydrogel for cartilage tissue engineering, which gels within 10–14 min and has excellent biocompatibility and regenerative properties [68]. In another interesting study, Fan et al. produced chitosan/hyaluronan SPAAC hydrogels that gel between 21 and 58 min, contingent upon the ratios of precursor polymers [69].

Chen et al. developed a hydrogel consisting of bio-orthogonally crosslinked hyaluronate and collagen for application in corneal stroma, as reported in recent publications. The hydrogel demonstrates potential as a vision-enhancing material by creating a transparent, smooth anterior curvature on corneal wound beds and allowing for control over this curvature during the gelling process. Therapeutic applications, including point-of-care settings and operating rooms, align effectively with the gelation time, which is quantified in minutes. The material effectively addresses stromal defects without the use of sutures and facilitates crosslinking in ambient water conditions, eliminating the need for light irradiation or additional triggers or initiators. Both in vitro and in vivo studies demonstrated that this hydrogel effectively supported epithelialization and exhibited significant cytocompatibility. The material’s ability to restore functional vision in cases of corneal abnormalities that threaten eyesight necessitates further investigation [70].

Liu et al. developed an innovative injectable bone cement system using organic-inorganic nanohybrids (click-ON) crosslinked by SPAAC click chemistry for biomedical purposes. RBMSC stem cells exhibited superior adhesion, proliferation, and differentiation on crosslinked click-ON scaffolds. A rat cranial defect model demonstrated that the click-ON device facilitates bone repair with little cytotoxicity. The click-ON cement exhibits reduced cytotoxicity, diminished heat production, and enhanced biodegradability compared to the clinically used poly (methyl methacrylate) (PMMA) bone cement. This injectable click-on cement technology has distinctive features that may enhance bone tissue engineering and many biological applications [71]. In another study, Zhou et al. synthesized stable crosslinked hydrogels for tissue engineering from isonitriles, chlorooxims, azomethine imines, and tetrazines using SPAAC-free copper click chemistry. The crosslinking of Isonitrile chlorooxime exhibited the most rapid gelation kinetics at physiological pH, devoid of gaseous byproducts, making it optimal for cell-encapsulating hydrogels. The stability and orthogonality of isonitriles, ligation, and SPAAC facilitated the sequential and selective functionalization of cell-encapsulating hydrogels with model small molecules. The results indicate that isonitriles ligations may be used in bioorthogonal procedures to develop cellular microenvironments, especially in multicomponent reactions for the fabrication of intricate hydrogel scaffolds in tissue engineering [72].

Furthermore, Chang et al. revealed that stoichiometric integration of cholesterol into azide-functionalized hyaluronic acid (HA) can selectively modify the microstructures and hydrophilic amphiphilicity of HA by SPAAC. The dynamic crosslinking of cholesteryl hyaluronic acid into physical gels with improved and recoverable viscosity transpires at low cholesterol concentrations due to hydrophobic interactions among cholesterol moieties. The chemical and physical interactions among cholesteryl HA crosslinks are illustrated by the crosslinking of residual azides by SPAAC. At elevated cholesterol concentrations, cholesterol-HA self-assembles into multilamellar nanoparticles (NPs) including an inner core comprising alternating layers of cholesterol and HA, with an outside layer that is hydrated and abundant in HA. Cholesterol HA with customized amphiphilicity can be utilized to create versatile macromolecular building blocks for regenerative medicine applications, such as viscoelastic physical gels, dual-crosslinked gels, and multilamellar nanoparticles [73]. Table 2, summarizes representative studies with details on hydrogel composition, click reaction type, and primary tissue-engineering applications, highlighting the advantages and limitations relevant to biomedical use.

### 3.2. Thiol-Ene Based Click Hydrogels

Alkene and thiol functional groups serve as the reactants in the thiol-ene click reaction. This method enables the formation of orthogonal networks because of its high efficiency, quick reaction rates, strong selectivity, lack of a necessary initiator, resistance to oxygen and moisture, and superior biocompatibility. The thiol-ene technique utilizes both step-growth and chain-growth polymerization methods. Following the generation of radicals, the initiator produces thiyl radicals by abstracting protons from the thiol groups of the reactants. Thiyl radicals subsequently produce carbon-based radicals, so commencing the creation of carbon–carbon double bonds. The created carbon-based radical may move through a carbon–carbon double bond, undergo chain transfer, or react with a new thiol group to produce a thiyl radical [80]. Thiol-functionalized compounds can covalently crosslink with carbon-carbon double bonds. Researchers are examining thiol-based reactions in click chemistry for the synthesis of polymeric materials, owing to their safety and non-toxicity in organic synthesis [81]. The thiol-ene and thiol-ene reactions, which need no toxic chemicals or catalysts, provide functional hydrogel biomaterials suitable for tissue engineering and regenerative medicine.

Granja’s group used thiol-ene photo-crosslinking to create a pectin-based hydrogel that is degradable by cells. The cell-instructive hydrogel system included norbornene-functionalized pectin, monocysteine cell-adhesive ligands for integrin interaction, and an enzymatically cleavable biscysteine peptide crosslink. Researchers used gel-encapsulated dermal fibroblast cells to investigate the formation of skin tissue in a laboratory setting. High-homogeneity thiol-norbornene hydrogel networks may facilitate tissue growth by establishing a specific environment. Furthermore, there was an absence of chains or homo-polymerization of strained norbornene groups in thiol-norbornene reactions [82]. In another study, Brown et al. introduced a dynamic, viscoelastic, photopolymerized hydrogel system including an 8-arm PEG thiol macromer, a thioester di (vinyl ether) crosslinker, a caged thioester catalyst, and norbornene-RGD for cellular adhesion. Hydrogel technology has shown efficacy in vitro when used to primary human mesenchymal stem cells. Thiol-esters may be extensively used in cellular scaffolding and has biological significance [83]. Furthermore, Hawker et al. discovered that a thiol-ene crosslinked PEG-based hydrogel system exhibits superior cohesion compared to traditional PEG hydrogels [84]. They used PEGDA and dithiothreitol to synthesize a thiol-ene-based hydrogel that exhibits glucose responsiveness and self-healing properties. This hydrogel is intended for brain tissue engineering. Upon the addition of glucose to this injectable hydrogel, it serves as a sacrificial substance, resulting in the formation of three-dimensional branching tubular structures. The primary matrix consists of components that are indestructible. Endothelial and neural stem cells produced vascularized neural tissue after 14 days [63].

Modifications to the norbornene group are often favored because to their rapid responsiveness and high safety for cellular applications [82]. Thiol-ene chemistry was recently used to fabricate cell arrays using poly (oligoethylene glycol methacrylate), cell-adhesive peptides RGD and REDV, together with alkene-linked residues (allyl and norbornene). The configuration of biomolecules inside the polymer enables the modulation of cellular adhesion. Human umbilical vein endothelial cells exhibited enhanced adhesion to structured surfaces using RGD-patterned polymers [85]. Sharma et al. integrated photo-click chemistry, microcontact printing, and electrospinning to develop a microarray technique using thiol-ene photo-clickable peptides for investigating cellular behavior in a three-dimensional microenvironment. Norbornene facilitated a photo-click reaction to retain peptides inside the fibrous matrix, while thiol-ene was used to conjoin electrospun fibrous structures. This approach enables the accurate incorporation of any thiol-containing reactant into the 3D build using free norbornenes. They used a standard contact printer to include cysteine-containing peptides into a microarray model to test the hypothesis on many more cell lines [86].

In addition, Liang et al. synthesized a composite hydrogel composed of silk fibroin (SF) and poly (ethylene glycol) diacrylate (PEGDA). A thiol-ene click reaction induced by UV light at 405 nm linked the glutathione on the silk fibroin with the acrylate moiety on the polyethylene glycol backbones. The crosslinked PEGDA/SF hydrogel effectively released rhodamine B due to its low swelling ratio and high biocompatibility in vivo [87]. Lueckgen et al. synthesized cell-empowered, enzymatically degradable, UV-initiated photo-crosslinked alginate hydrogels using norbornene-modified alginate, a photo initiator, and two thiol-coupled peptides (an RGD peptide and a degradable crosslinker) [88]. Despite the similarity in reaction mechanisms, thiol-ene reactions are more probable with electron-rich alkenes than thiol-Michael additions. Thiol-ene click chemistry may occur in physiological aqueous solutions with rapid photopolymerization kinetics, making it ideal for 3D bioprinting [89]. Recent research revealed that a pH- and Thermoresponsive chitosan/PNIPAM hydrogel system, using UV-initiated thiol-ene crosslinking, exhibited over 95% compatibility with human mesenchymal stem cells after 24 h [90]. Zhou et al. used maleic chitosan, thiol-linked PVA, and a biocompatible initiator to investigate a photo-polymerized thiol-ene hydrogel for tissue engineering applications. This hydrogel exhibits excellent compatibility with fibroblasts and has superior mechanical characteristics [91].

Employing azide-alkyne click chemistry, Takemoto’s team has conjugated hydrogels with biological cells and tissues. They incorporated an alkyne group into alginate and modified live cells and tissues with an azide group to target sialic acid residues on the cell surface. We cultivated azide-modified cells with alkyne-modified hydrogels to synthesize gels. Live cells based on hydrogel exhibited a higher likelihood of survival [92]. Ding and his colleagues demonstrated that a clickable, combinatorial system might enhance the contractility of human smooth muscle cells in vascular tissue creation. The fibrous hydrogel system, characterized by various forms, compositions, and mechanical stiffness, was used to replicate smooth muscle. They used hydrogels and protein array technology to illustrate that pharmaceuticals may influence smooth muscle contractility in vitro [93].

In accordance with recent research, Afonso et al. employed alginate, gelatin, and carboxymethyl cellulose, incorporating norbornene and thiol groups for optimal substitution. These modifications enhanced polymer reactivity and facilitated thiol–ene click reactions to produce orthogonally photo-crosslinked hydrogels. The equimolar combination of these polymers with AuNR colloids resulted in improved bioprinting, cell culture, and biosensing inks. Chemically stable hydrogels incorporating diverse polymers and gold nanorods (AuNRs) serve as SERS substrates for the detection of physiologically significant chemicals. They evaluated the rheology, swelling, degradation, microstructure, and biocompatibility of the hydrogels to identify appropriate plasmonic inks for 3D-printed scaffolds [94].

In another interesting research by Bailey et al., thiol-ene click chemistry is suitable for investigating how modifications to HPMC’s hydrophobic, steric, or pi stacking features might be utilized to modify PNP hydrogel characteristics. PNP hydrogels can be synthesized with consistent network density and diffusion, while exhibiting varied in vivo retention durations, so offering a novel method for regulating therapeutic cargo release. This article outlines a straightforward thiol–ene click derivatization process for modifying PNP hydrogels using HPMC. This technique significantly enhances research on polymer–nanoparticle interactions due to the plethora of commercially available thiol-containing compounds and cysteine-containing peptides. Rheometry demonstrated that the identification of the HPMC pendant group could significantly alter the stiffness, flow, and recovery characteristics of bulk materials. Adjustable mechanical properties may be employed to modify hydrogel retention times in vivo, illustrating the potential for customized therapeutic delivery [95].

Moreover, Atmani et al. demonstrated the straightforward modular synthesis of water-soluble allyl-functionalized polysaccharide carbamates (AFCs). The thiol–ene reaction chemically crosslinked AFC derivatives into hydrogels within seconds of exposure to UV light at 365 nm. The gels were thoroughly characterized and exhibited significant mechanical strength, swelling capacity, porosity, and a uniform pore structure. AFCs were optimal for 3D printing because of their rapid gelation and superior properties. This will enhance biomedical applications for biobased polymers. The synthesis of isotropic gel materials using partial UV irradiation is a simple and intriguing technique. Extensive investigations on 3D printing equipment are now being conducted to enhance process parameters and printability [96].

Oxidizing agents and neutral pH facilitate the oxidation of two thiol groups, resulting in the formation of disulphide bonds [97]. While oxidizing agents may enhance the kinetics of disulphide crosslinking gelation, they remain somewhat slow relative to other click chemistry methods [98]. Gyarmati et al. discovered that disulphide crosslinked hydrogels underwent a transition from liquid to gel within 6–24 h across various pH solutions and within 10–1000 s in the presence of chemical oxidizing agents (cystamine and 3,3′-dithiodipropionic acid, respectively), exhibiting no degradation for up to one week in PBS buffer and one week in vivo [99].

Utilizing activated disulphide (pyridyl disulphide) to synthesize hydrogels by disulphide exchange processes, rather than thiol oxidation, might expedite gelation; nevertheless, it produces tiny molecular byproducts that may be detrimental in tissue engineering. This crosslinking technique is beneficial as the disulphide bonds respond to reductive environments, characterized by elevated concentrations of reducing agents (e.g., glutathione or dithiothreitol) and/or marginally reduced local pH levels, commonly found at various disease or infection sites [100]. This facilitates improved degradation for precise medication or cell delivery to a specific location. Disulphide linkages facilitate the redox-responsive degradation of smart hydrogels in the presence of glutathione and other reducing agents [101]. Polymers containing thiol groups interact with natural thiols, disulphide, and electrophilic agents. Disulphide crosslinking is not bio-orthogonal; nevertheless, accelerating the process using previously published techniques reduces the likelihood of thiol cross-reactivity with native functional groups, hence facilitating “kinetic bio-orthogonality” in practice [102].

Direct encapsulations of cells inside disulfide-bonded hydrogels are also effective. In vitro growth of disulphide crosslinked hyaluronan hydrogels sustained L-929 murine fibroblasts for 3 days, while many cell types, including fibroblasts, endothelial cells, and mesenchymal stem cells, exhibited viability after 7 days (as assessed by a live-dead test) [103]. Disulphide crosslinking may be appropriate for secondary crosslinking of in situ gelling hydrogels in tissue engineering applications. The slow disintegration of the oxime bond and multi-arm PEG under physiological conditions inhibits cells from reconstructing the scaffold after 7 days, hence inhibiting cell spreading. The degradability of the polymer backbone may be crucial for tissue regeneration using oxime crosslinks. Table 3 outlines representative compositions, reaction setups, and biomedical applications, illustrating the versatility of thiol–ene chemistry for creating injectable, 3D-printable, and biofunctional scaffolds with tunable mechanical and biological properties.

### 3.3. Boronated Ester Click Chemistry

Boronic acid interacts with a diol to produce boronate ester bonds. Choosing the appropriate boronic acid may facilitate rapid gelation at physiological temperature and pH levels (pH 8–9). This exothermic reaction occurs rapidly [107,108]. Amaral et al. discovered that the amalgamation of phenylboronic acid-functionalized β-glucan laminarin with poly (vinyl alcohol) resulted in the rapid formation of a boronate ester-crosslinked hydrogel under physiological conditions [109]. Boronate esters rapidly decompose under acidic and hydrogel conditions. Boronate ester click chemistry may degrade gel in response to biological signals by substituting the diol with naturally occurring carbohydrates with cis-diol groups that exhibit superior binding to the boronic acid moiety [109]. Certain boronate ester molecules persist for three months in pure water, whereas others decompose within minutes under physiological conditions [110,111]. Boronate ester gelation occurs rapidly and has hydrolytic stability, making dual-crosslinked hydrogels extensively used. Wu et al. synthesized a dual-crosslinked hydrogel using complementary diol and benzoxazole-based monomers, together with traditional boronate ester crosslinking featuring a nopoldiol-based benzoxaborolate, a cyclic hemiboronic acid with a lower pKa than standard arylboronic acids. The dual-crosslinked hydrogel persisted for 10 to 20 days in an acidic pH environment, but the conventional boronate ester bonds between the arylboronic acid and a diol deteriorated after 120 min upon exposure to a polyol solution [111]. Liu et al. synthesized dual-crosslinked hydrogels using boronate ester and hydrazone linkages. Dual-crosslinked structures exhibited greater strength and durability under physiological settings, but single-crosslinked gels gelled more rapidly (within 2 h) [112]. Boronate esters are optimal for double crosslinking and for scaffolds that are weak or rapidly deteriorating. Encapsulating cells with boronate esters demonstrate cytocompatibility. Hydrogels composed of hyaluronic acid, a boronic acid derivative, and 1-amino-1-deoxy-D-fructose demonstrated over 80% fibroblast cell viability after 7 days, but those formulated with phenylboronic acid-functionalized β-glucan laminarin and poly (vinyl alcohol) preserved preosteoclast vitality for 48 h. one hundred eighty Hydrogels crosslinked with boronate esters have tunable viscoelastic characteristics owing to the dynamic nature of the boronate ester bond, which responds to stress or strain via reversible crosslinking and de-crosslinking, enabling researchers to investigate time-scale-dependent mechano-transduction in cells [31].

To augment the anticancer efficacy, Jung et al. devised a sophisticated thermos-responsive injectable hydrogel derived from a copolymer including boronic acid and consisting of poly (ε-caprolactone-co-lactide)-b-poly (ethyleneglycol)-b-poly (ε-caprolactone-co-lactide) (PCLA). The results indicate that PCLA copolymers are promising as injectable depots and suitable carriers for the targeted delivery of DOX in the treatment of hepatocellular carcinoma [113]. In another study, Wang et al. produced a self-healing, conductive SPPM-h 3D printing hydrogel ink utilizing 3-aminophenyl boronic acid (PBA)-grafted sodium alginate (SA) and polyvinyl alcohol (PVA) with MXene. The physicochemical properties of hydrogels and the dynamic self-healing caused by boronic ester bonds were thoroughly investigated. The SPPM-h 3D printing hydrogel ink exhibits enhanced tensile properties and electrical conductivity (0.034 S/m) relative to the PSM-h double-network hydrogel (0.007 S/m). SPPM-h has remarkable self-healing capabilities, fully recuperating within 30 s after fracture. Rheological analyses indicate the substantial recovery of SPPM-h. As a consequence, SPPM-h is a promising 3D printing ink for conductive, self-healing network architectures [114].

Furthermore, Miki et al. developed two FBA-HA/Gluca-HA systems and demonstrated that the system with the highest degree of substitution (pH 3.90) rapidly gelled in a buffer solution at physiological pH (pH 7.4). The in situ gelation of the HDS was unaffected by the presence of 5.5 mM Glc in the buffer solution into which the FBA-HA/Gluca-HA system was introduced, despite its competition with boronate ester crosslinking. This makes it a viable clinical contender. The boronate ester-crosslinking sites of the polymers increase markedly when pH transitions from acidic to physiological, leading to the creation of a resilient gel. The study revealed that pH-responsive in situ gelation necessitated elevated degrees of substitution (DS) and potassium (K), with Gluca serving as an advantageous cis-diol-containing moiety for BA derivative-modified polymers. The FBA-HA/Gluca-HA combination, which rapidly gels under physiological conditions, may enhance drug delivery, cell engineering, and wound dressings [115]. Recently, Miki’s research group on injectable self-healing hydrogels discovered that salicylic acid serves as a possible diol moiety for boronate ester crosslinking. In situ gelling materials that rapidly form gels upon contact with body fluids are compelling for medicinal applications. Hydrogels that are flexible and pH-responsive may be synthesized using boronate ester crosslinking. Phenylboronic acid (BA) derivatives and glucamine-modified hyaluronan were used to manufacture an acidic system. By means of a condensation procedure, 3-amino-4-fluorophenylboronic acid and D-glucamine were converted into sodium hyaluronate to produce mixed modified hyaluronan systems. Additionally investigated the influence of pH on the rheological properties of the modified hyaluronan system [116].

Table 4 highlights key polymer compositions, phenylboronic acid (PBA)–diol dynamic crosslinking strategies, and their applications in regenerative medicine, including self-healing scaffolds, glucose-responsive systems, injectable bone and cartilage matrices, and multifunctional wound-healing platforms. Advantages such as reversible bonding, stimuli-responsiveness, and biocompatibility are noted alongside emerging translational opportunities.

### 3.4. Diels–Alder Click Chemistry

Diels–Alder click chemistry reactions typically entail a diene and an alkene without the necessity of a coupling agent and catalyst. Aqueous DA reactions, like to click chemistry reactions, are rapid, adaptable, selective, and efficient [124,125]. They generate no deleterious byproducts. This dependable method is extensively employed to produce crosslinked hydrogels for tissue engineering [17,62]. There are several research on DA click crosslinked hydrogel for tissue engineering. For instance, Nimmo et al. employed the Diels–Alder process to crosslink hyaluronic acid-based furan-modified hydrogels with di-maleimide-linked polyethylene glycol. They suggested altering the proportions of furan and maleimide to improve mechanical properties and degradation. Hydrogels were assessed for cytocompatibility with human epithelial cells and their degradation characteristics [126].

In another study, Wang et al. synthesized HA/PEG hydrogel with injectable characteristics using a dual-crosslinking procedure. The injectable HA/PEG hydrogel may rapidly gel within 30 s through a photo-crosslinking reaction between HA-Furan and LAP, which preserves encapsulated cell viability and facilitates clinical application. The thermally generated DA click chemistry progressively transpired between Furan and Mal groups, which incrementally enhanced the crosslinking density and mechanical characteristics of the hydrogel. The injectable HA/PEG hydrogel effectively contained the ATDC-5 cells in situ and demonstrated excellent cytocompatibility. The aforementioned results indicate that the injectable hydrogel, exhibiting progressively enhanced mechanical characteristics through a photo-crosslinking reaction and thermally induced DA click chemistry, holds significant promise for applications in cartilage tissue engineering [127].

Koehler et al. synthesized DA-based hydrogels for drug release and osteogenic differentiation utilizing the Diels–Alder reaction and maleimide PEG macromer and furan dexamethasone peptide. Hydrogels increased mineralization and alkaline phosphatase activity by sixfold [128]. Moreover, Smith et al. showed that methylfuran-maleimide click-crosslinked hydrogels can be synthesized efficiently at pH 7.4, enabling cell encapsulation and 3D culture. They also showed that replacing furan with the electron-rich methylfuran increased the rate of the conventional furan-maleimide reaction [129].

In addition, Shoichet’s group identified that Diels–Alder-synthesized furan-modified hyaluronic acid and bis-maleimide polyethylene glycol hydrogels could spatially sequester biomolecules (galactose) within a two-photon laser photo pattern. They elucidated the effects of cryo-gelation and thawing temperature on porosity and pore dimensions. The substitution of furan modified the mechanical properties of the hydrogel [130]. Bai et al. developed dual crosslinked injectable self-reinforcing hydrogels for tissue engineering. Dual crosslinking utilizes cyclodextrin, adamantane, PIPAM, and DA-based click chemistry. Dual crosslinking enhanced hydrogels, and in vivo experiments demonstrated bone regeneration in the absence of cells or growth hormones. Employing DA-based click chemistry, researchers developed a dual-crosslinked injectable chondroitin sulfate hydrogel incorporating bone morphogenetic protein-4 to address cranial anomalies in rats. Both noncovalent and covalent methods were employed to crosslink ChS-Furan, F127-linked maleimido, and PEG-AMI hydrogels. Rats injected with BMP-4 generated new bone after 12 weeks [131]. Fan et al. created biodegradable hydrogels for adipose tissue engineering employing DA-based click chemistry with furan-linked and maleimide-linked hyaluronic acid. Human adipose-derived stem cells were assessed for cytocompatibility within hydrogels. Hydrogels promoted stem cell proliferation without triggering differentiation [132].

Furthermore, Madl et al. found that fulvenes and maleimide dienophiles quickly crosslink multi-arm poly (ethylene glycol) (PEG) hydrogels and hybrid PEG-engineered protein hydrogels with better cell encapsulation stability [133]. Lu et al. used self-healing and injectable hydrogels for in vivo regeneration of cerebral bone. Characterization of catechol-modified N-(furfural) chitosan (CFC) dual crosslinked hydrogels for tissue engineering utilizing coordination bonds and Diels–Alder click chemistry. Iron-catechol coordination regulates crosslinking and self-healing. Dual crosslinking enhanced the mechanical properties of the hydrogel [134,135]. Abandansari et al. developed a compelling interpenetrating hydrogel system (furan-linked gelatin and maleimide-linked PEG via DA click chemistry) with in situ gel formation, thermosensitivity (chitosan-Pluronic F127), high mechanical properties, and biocompatibility for cardiac cell retention Puran-maleimide equivalents improve crosslinking, gelation, and mechanical characteristics. Changing the PEG/gelatin ratio changes hydrogel mechanical characteristics [136].

Yu et al. synthesized cartilage utilizing tissue-adhesive DA click chemistry hydrogels. Dimaleimide PEG, furan adipic dihydrazide, and furan aldehyde were incorporated into HA to create the double-crosslinked network hydrogel. Hydrogels enhance mechanical properties, swelling capacity, self-repairing abilities, and cartilage adhesion [125]. Bai et al. designed a triple-crosslinked injectable hydrogel for the repair of cranial bone. The hydrogel was composed of modified sodium alginate, bioglass, and chondroitin sulfate. Triple crosslinking was achieved by noncovalent interactions, acylhydrazone linkages, and DA-based click chemistry. The hydrogel formulation promoted bone repair in vivo owing to its physicochemical characteristics [137].

Retro-DA chemistry has the potential to compromise crosslinks. The degradation of some DA-crosslinked hydrogels is postponed under physiological circumstances. DA-crosslinked chitosan-based hydrogels exhibited a 98% cell viability after 21 days in vitro, whereas Pluronic F127/Gelatin/PEG demonstrated over 95% viability. Uncrosslinked chitosan had an 82% degradation in lysozymes and a 47% degradation in PBS after 14 days [136]. Conventional Diels–Alder reactions employ electron-rich dienes and electron-poor dienophiles. Conversely, IEDDA chemistry facilitates hydrogel formation at physiological pH and temperature [41,42,138]. IEDDA-crosslinked hydrogels are cytocompatibility and effective for peptide immobilization, whereas those with phenyl tetrazine activity and dienophile have increased thermodynamic stability due to the nitrogen-generated crosslink’s irreversibility [139,140,141].

### 3.5. Oxime and Hydrazone Formation

In oxime click chemistry, amino-oxy groups undergo reactions with aldehydes or ketones. Rapid reactions interact with biomolecules and cells in an orthogonal manner. Water is generated in the absence of a catalyst, UV light, or external temperature [24]. Physiologically stable imide hydrazone and oxime linkages are produced by reactions [142]. Peptides, proteins, and DNA undergo modification by the oxime process due to its superior stability compared to thiol groups [143]. Based on recent reports, researchers have demonstrated the superiority of oxime click chemistry over thiol-based crosslinking strategies. Oxime click chemistry provides higher chemo selectivity, greater stability under physiological conditions, and excellent biorthogonality compared to thiol-based linkages, which are often prone to oxidation and require radical initiators. In tissue engineering, oxime-based hydrogels enable mild, initiator-free gelation with dynamic covalent bonds that impart self-healing and adaptability, thereby outperforming thiol–ene systems that rely on photo initiators. Recent advances in both sensor-responsive hydrogels and tissue engineering further highlight the versatility of oxime chemistry for multifunctional biomedical applications [144,145]. Their applications encompass polymer-protein ligation, cell surface modification, and in vivo tissue labeling [146]. Gelation may occur at physiological or near-physiological pH, although slightly acidic circumstances expedite it. PEG hydrogels synthesized by oxime bond formation gelled after 6 min at pH 6 but took 30 min at pH 7.2, while hydrogels made by mixing alkoxy-amine and aldehyde-functionalized alginate precursor polymers gelled faster from 4 to 50 °C. Oxime bonds tolerate hydrolysis better than most click crosslinks [142,147,148].

Boehnke et al. developed tunable degradation hydrogels utilizing a PEG-based imine hydrogel platform that integrates the biocompatibility and stability of oxime bonds with the reversibility of hydrazone bonds. The facile functionalization of PEG permits the modification of these gels with hydrazide and hydroxylamine groups. The amalgamation of oxime and hydrazone chemistries enhanced hydrogel stability from 24 h to 7 days. High cell viability was observed with covalently attached RGD peptides. This system is applicable for cell delivery and tissue engineering in both research and clinical contexts, as it allows for the modulation of hydrogel mechanical properties and degradation while encapsulating cells [149].

Grover et al. employed 8-arm amino-oxy-PEG, RGD peptides, and glutaraldehyde to create oxime click chemistry-derived hydrogels for live MSC encapsulation. The researchers hypothesized that amino-oxy groups forming persistent oxime connections in aqueous settings stabilize hydrogels better than amine groups forming imine bonds [40]. Sanchez-Moran et al. incorporated alkoxyamine (AA) functional groups into sodium alginate (NaAlg) polymers. The novel alginate hydrogels’ dual crosslinking (ionic and oxime) enabled the formation of viscoelastic hydrogels in microbead or microthread configurations. Encapsulating and cultivating 2PK-3 cells within these hydrogel beads demonstrated biocompatibility, rendering this innovative material platform promising for biomedical applications requiring biomimetic viscoelastic microenvironments in a defined shape [148].

Hardy et al. explored amino-oxy modified PEG and aldehyde-modified HA-based oxime-crosslinked biodegradable, biocompatible hydrogels for soft nerve tissue engineering with adjustable mechanical properties. Hydrogel cell adhesion was examined using human MSCs, whilst Schwann cells were employed for cytotoxic tests. Both demonstrated potential for nerve tissue regeneration upon the incorporation of collagen I into hydrogels [150].

As an interesting study, Baker et al. conducted a compelling study that identified an injectable oxime-crosslinked hydrogel as a biomimetic substitute for the vitreous body. Subsequently, they developed a hyaluronan-based hydrogel crosslinked via oxime chemistry to replicate the vitreous body. This system involved the modification of hyaluronan with aldehyde and ketone groups, which were then crosslinked with poly (ethylene glycol)-tetraoxyamine. This approach facilitated tunable gelation kinetics and yielded a hydrogel exhibiting optical and physical properties akin to natural vitreous. The hydrogel demonstrated minimal swelling, could be administered using fine needles, and was safe for retinal photoreceptors in vitro. In rabbit models, the oxime-crosslinked hydrogel preserved intraocular pressure, retinal structure, and function for up to 90 days before biodegradation commenced around day 28. These results underscore the potential of oxime chemistry in the development of injectable, stable, biodegradable vitreous substitutes with superior biocompatibility, thereby advancing clinically applicable materials for retinal repair [147].

A biorthogonal oxime click chemistry-derived hydrogel for 3D cell culture was described by Farahani et al. This work generated a cell-encapsulated hydrogel utilizing moderate UV light photopolymerization. The photo-mediated oxime reaction immobilized proteins with alkoxyamines in UV-exposed areas. Sample sites are polymerized by photo-mediated oxime reactions. All UV-irradiated polymer solution gelled except photo-masked portions [53]. Hentzen et al. crosslinked model collagen peptides using 4-oxoacetamido-proline and 4-aminooxy-proline through oxime linkages. The covalently linked peptides established stable collagen triple helices [151]. Tamura et al. established a protein labeling affinity-guided oxime catalytic system using pyridinium oxime and N-acyl-N-alkyl sulfonamide. This approach can selectively identify natural proteins in test tubes and cell lysates under standard physiological conditions. Employ this catalytic system to fluorescently label and visualize membrane proteins in living cells. This system was tested on slices of mouse brain [152]. Table 5, compiles advances in oxime- and hydrazone-crosslinked hydrogels for tissue engineering, highlighting their mild gelation conditions, tunable mechanics, self-healing properties, and versatility across applications such as vitreous replacement, wound healing, anti-adhesion barriers, and MSC scaffolds.

The straightforwardness, specificity, and water compatibility of dynamic covalent chemistry for hydrazone and oxime connections make it attractive. Wang et al. examined the influence of ions on the kinetics of hydrazone and oxime reactions. Researchers discovered that fundamental salts such as sodium chloride (NaCl) may stabilize the rate-limiting transition state and enhance oxime production at neutral pH with aldehyde and less reactive keto substrates, hence accelerating dehydration. NaCl or divalent salts such as MgCl_2_ or CaCl_2_ may enhance reaction rates for these procedures. Under physiologically relevant circumstances, small molecules and multifunctional biomacromolecules enhanced reaction rates in arylhydrazone and oxime-based hydrogels. This straightforward technique for altering the kinetics of hydrazone and oxime reactions will enhance the development of biomaterials and bioconjugation [156].

### 3.6. Pseudo-Click Hydrogels

Thiol-Michael pseudo click chemistry creates thioester bonds by adding a thiol group to a vinyl sulfone, acrylate, or maleimide double bond with or without a catalyst [157]. Advantages include tolerance to different functional groups, enhanced accessibility to thiol and ene-modified reagents, and faster reaction rates [158]. Hubbell’s group synthesized crosslinked step growth hydrogels using Michael addition [159]. Thiol-linked peptides, dithiol-linked PEG, and multi-acrylate-linked PEG were reactants. Over the last decade, many global organizations have developed Michael-type addition reaction hydrogels. This method combined chitosan-PEG, fibronectin, hyaluronic acid, and gelatin-PEG to make hydrogels [160]. Young and Engler crosslinked thiol-HA and PEGDA hydrogels for cardiac tissue engineering using Michael addition [161].

Recently, Pupkaite et al. demonstrated that injectable hydrogels based on extracellular matrix-derived polymers show much promise in the field of tissue engineering and regenerative medicine. However, the hydrogels reported to date have at least one characteristic that limits their potential for clinical use, such as excessive swelling, complicated and potentially toxic crosslinking process, or lack of shear thinning and self-healing properties. They hypothesized that a collagen hydrogel crosslinked using thiol-Michael addition click reaction would be able to overcome these limitations. To this end, collagen was modified to introduce thiol groups, and hydrogels were prepared by crosslinking with 8-arm polyethylene glycol-maleimide. Rheological measurements on the hydrogels revealed excellent shear-thinning and self-healing properties. Additionally, only minimal swelling (6%) was observed over a period of 1 month in an aqueous buffer solution. The reported thiolated-collagen hydrogel crosslinked using thiol-Michael addition click reaction overcomes most of the challenges in the injectable hydrogel design and is an excellent candidate for cell delivery in regenerative medicine and tissue engineering applications [162].

In hydrogel formation, thiol–acrylate and thiol–maleimide linkages are commonly used to establish durable, covalently bonded networks with tunable stiffness and gelation profiles. The adaptability of this method allows for precise regulation of hydrogel properties, including degradation behavior, functionalization level, and biological responsiveness. Due to its biocompatibility and reaction specificity, Michael addition is especially suitable for developing injectable hydrogels, in situ forming gels, and ocular bioadhesives where controlled crosslinking is essential. Subsequently, Tortora et al. present a pioneering approach for the formation of poly (vinyl alcohol) (PVA)-based hydrogels using the Michael addition strategy, specifically designed for safe in situ vitreous replacement. Recognizing the risks associated with radical initiators in conventional crosslinking methods, the authors devised a two-component system: methacryloyl-modified PVA and thiol end-capped PVA, synthesized via selective oxidation and cysteine conjugation. Upon mixing in physiological conditions, these macromers undergo a Michael-type nucleophilic addition, forming a stable hydrogel network without the need for radicals or photo initiators. The resulting hydrogels exhibit tunable gelation times compatible with surgical requirements, as well as favorable viscoelastic and swelling properties that can be tailored by adjusting the macromer composition. This strategy not only minimizes potential cytotoxicity but also enables precise control over network structure and mechanical characteristics, making these PVA hydrogels highly promising for biomedical applications such as vitreous substitutes, where biocompatibility, transparency, and mechanical integrity are critical [163].

Instead of acrylate, researchers used highly reactive vinyl sulfone in thiol-Michael reactions due to its strong links and electron-withdrawing properties [164]. Patterson and Hubbell created tissue-application thiol-Michael reaction-based PEG hydrogels using 4-arm PEG coupled with vinyl sulfone, cell-adhesive peptides, and matrix metalloproteinase-degradable peptides Jin et al. synthesized collagen and chondroitin sulfate in chondrocyte-cultured HA hydrogels via thiol-Michael addition. For cartilage regeneration, they used thiol-conjugated hyaluronic acid and tetravalent polyethylene glycol with vinyl sulfone [165]. In thiol-maleimide PEG hydrogels, fibronectin and RGD peptide increased human fibroblast adhesion and proliferation. The same research group created a glutathione-sensitive hydrogel using the thiol-Michael reaction. At 4 °C and pH 6.6, gel formation was suppressed. Temperature and pH lowering helped vortex mixing and sample input [166]. Bang et al. created chondroitin sulfate-gelatin dual crosslinked hydrogels to increase cell adhesion. The researchers used the Michael-type click chemical reaction and N-(3-diethylpropyl)-N-ethylcarbodiimide hydrochloride to produce hydrogels and evaluate them for drug delivery and tissue engineering [167].

The thiol-Michael addition process can be initiated by several catalysts, including bases, metals, organometals, Lewis’s acids, and nucleophiles. Among these catalysts, bases and nucleophiles have the highest efficiency, demonstrating minimal propensity for side reactions. Both base- and nucleophile-catalyzed thiol Michael reactions occur swiftly, achieving high conversion rates under moderate circumstances; nevertheless, their processes exhibit subtle variations. The base extracts a proton from the thiol, producing a thiolate anion that participates in the thiol-Michael addition process. The nucleophile interacts with the electron-deficient double bond, forming an intermediate carbanion that subsequently deprotonates the thiol, yielding another thiolate anion. Under physiological settings, Liu et al. synthesized an in situ thiol-Michael hydrogel utilizing glycidyl methacrylate-modified dextran backbones crosslinked by dithiothreitol. The mechanical characteristics, gelation time, and swelling behavior of the hydrogel exhibited significant pH dependence. Moreover, 3D cell encapsulation experiments demonstrated elevated viability of MSC and NIH/3T3 fibroblasts [168].

Aldehyde-hydrazide pseudo reactions, like other click chemistry reactions, are straightforward, adaptable, and reversible, yielding non-toxic end products. Numerous polysaccharides were treated with aldehyde and ADH derivatives to produce hydrogels. Bulpitt and Aeschlimann synthesized the inaugural hydrogel utilizing HA-ADH and HA-aldehyde [169]. Aldehyde-hydrazide crosslinked hyaluronic acid-based hydrogels facilitated the healing of rat brains, as demonstrated by Tian et al. [170]. This approach shows significant potential for the development of biomaterials for tissue regeneration. In situ crosslinkable hydrogel based on alginate and hyaluronic acid for cardiac tissue engineering by Dahlmann et al. Hydrogels generated contractile cardiac tissue utilizing rat heart cells [171].

In addition, Chang et al. presented the design and evaluation of an in situ-forming zwitterionic hydrogel for long-term vitreous substitution, utilizing the Michael addition strategy for crosslinking. The hydrogel is synthesized from a zwitterionic copolymer, poly (MPDSA-co-AC), which combines sulfobetaine methacrylamide for ultra-low biofouling and biocompatibility, and acryloyl cystamine to introduce thiol groups. Crosslinking occurs via a thiol–ene Michael addition reaction with α-PEG-maleimide as the crosslinker, enabling rapid in situ gelation under physiological conditions. The resulting hydrogels exhibit desirable properties for ocular implantation, including high transparency, controllable gelation time, tunable mechanical strength, and low swelling. In vivo studies in a rabbit model demonstrated that hydrogels with a thiol–ene ratio of 2:1 formed stable, space-filling, and transparent implants that remained biocompatible and intact for at least two months. This work highlights the advantages of the Michael addition strategy for hydrogel crosslinking, offering a promising, minimally invasive approach to create injectable, long-lasting, and biocompatible vitreous substitutes and potentially other soft tissue replacements [172].

Martinez-Sanz et al. created an injectable hydrogel composed of BMP-2-loaded HA for in vivo bone tissue development. These stable gels were synthesized using amidation and selective oxidation. Gelation occurred in 30 s, while BMP-incorporated hydrogel samples were cured at ambient temperature for 3 h. In vivo investigations utilized cured samples [173]. Numerous researchers have documented aldehyde-hydrazide hydrogels for tissue engineering, using PVA, elastin-like protein-HA, poly(N-isopropylacrylamide) (PNIPAM), HA, and HA combined with growth factors for bone tissue engineering [13,174,175,176]. Huang and Jiang have shown that amino-yne click chemistry may produce pH-sensitive hydrogels for localized drug delivery and tissue engineering. Their carboxymethyl chitosan-PEGDA hydrogels, which are pH-responsive, degradable, and injectable for tissue engineering and biomedical applications, did not necessitate a catalyst or initiator under physiological circumstances. Additionally, breakdown transpires to generate aldehyde and amine groups [13,174,175,176] at low pH, finally culminating in the total release of the medication at pH 2 [177]. This injectable amino-yne based click chemistry hydrogel is straightforward, cost-effective, and spontaneously forms a gel, potentially offering essential biocompatible hydrogels for tissue engineering applications without the need of hazardous catalysts or photo initiators.

### 3.7. Schiff Base Reaction

Schiff base chemistry is a technique commonly utilized to form dynamic covalent bonds in hydrogel systems. It is distinguished for its exceptional biocompatibility, reversibility, and uncomplicated reaction mechanism. The formation of imine bonds (–C=N–) occurs through the condensation of primary amines with aldehydes or ketones. These processes are especially beneficial for biomedical applications, such as ocular adhesives, injectable gels, and tissue engineering matrices, as they occur effectively in aqueous environments under moderate conditions, obviating the necessity for harmful catalysts or initiators. Hydrogels can dynamically adjust to their biological environments owing to the exceptional pH sensitivity and self-healing characteristics of Schiff base linkages. Imine bonds are often unstable in acidic conditions and prone to hydrolysis over time, unlike more robust covalent bonds such as oximes. Schiff base-crosslinked hydrogels are exceptionally appropriate for targeted drug delivery, corneal regeneration, and ocular tissue repair owing to their extensive polymer compatibility, customizable mechanical properties, and rapid gelation [178,179,180].

Chen’s team created a unique biocompatible polysaccharide-based self-healing hydrogel, composed of N-carboxyethyl chitosan, adipic acid dihydrazide, and oxidized sodium alginate. The outcomes of rheological recovery testing, macroscopic observation, and beam-shaped strain compression measurement demonstrated exceptional self-healing capability with a high healing efficiency (≈95%). Furthermore, NIH3T3 fibroblast cells were enclosed during gel formation, preserving good viability and proliferative potential [181]. Yin et al. formulated injectable hydrogels inspired by mussels, incorporating hydrazide-modified poly (L-glutamic acid) and dual-functionalized alginate (catechol- and aldehyde-modified alginate). The hydrogels demonstrated effective self-healing properties, excellent cytocompatibility, and robust bio-adhesion [182]. Furthermore, the Schiff base HA hydrogel serves as a delivery platform for growth factors and facilitates the adhesion and dissemination of MSCs when combined with integrin-specific fibronectin fragments [183].

Chand et al. created a dual-crosslinked interpenetrating network (IPN) hydrogel using Schiff base chemistry and photo crosslinking to fabricate biomimetic corneal stroma equivalents. Gelatin methacryloyl (GelMA) and oxidized carboxymethylcellulose (OxiCMC) create a hydrogel that enables primary crosslinking via the formation of dynamic imine (Schiff base) connections between their aldehyde groups and the amine groups of GelMA. UV-induced photopolymerization of GelMA guarantees network stability. This dual crosslinking method produces hydrogels with an improved compressive modulus (106.3 ± 7.7 kPa), significant optical clarity, and superior printability for DLP 3D bioprinting. The engineered structures exhibited favorable porosity, interconnected networks, superior cell viability (>93%) of encapsulated human corneal keratocytes, and robust structural integrity and reproducibility. The technique eliminates synthetic polymers and detrimental crosslinkers, ensuring ocular tissue creation is safe, biocompatible, and structurally sound [184].

Table 6 summarizes current advancements in Schiff-base crosslinked hydrogels for tissue engineering, highlighting their injectable, self-healing, and bioadhesive characteristics, as well as their applications in wound healing, cartilage and bone regeneration, and drug administration.

As illustrated in Figure 3, the diverse functional groups and click chemistry reactions employed for designing injectable hydrogels. Reactive moieties such as boronic acid, thiol, azide, alkene, aldehyde, and amine can undergo bio-orthogonal reactions, including azide-alkyne cycloaddition, Schiff-base formation, oxime ligation, boronate ester linkage, thiol–ene coupling, and Diels–Alder reactions. These chemistries enable tunable crosslinking, biocompatibility, and injectability, making hydrogels highly adaptable for biomedical applications [124].

## 4. Applications in Tissue Engineering

Hydrogels have found several applications in fields such as cell culture, medication delivery, tissue regeneration, and more. Hydrogel preparation relies heavily on “click” chemistry. Here, we look at some of the many uses for hydrogels made using “click” chemistry, including Cartilage and bone regeneration, Skin and wound healing, Neural and spinal cord scaffolds, Ocular and corneal regeneration, and Bioprinting and 3D fabrication. In the realm of biomedicine, regenerative medicine has paramount importance. The injectability, controlled release, biocompatibility, and degradability of biomaterials used for tissue regeneration, often hydrogels, are of the utmost importance. Researchers have often opted for “click” chemistry when developing hydrogels for tissue regeneration due to the difficulty of achieving a broad chemical crosslinking approach (Figure 4) [126].

### 4.1. Cartilage and Bone Tissue Engineering

The body cannot repair damage to cartilage tissue, leading to a lasting condition. Joint injuries may be effectively managed by the rapid development and placement of cells sourced from bone tissue [189]. Erlane and colleagues indicate that hydrogels hinder bone cell development in tissue regeneration because they do not contain sufficient oxygen molecules. Oxygen generators of the traditional kind carried risks and were prone to malfunctions. Consequently, “click” chemistry was employed to restore bone tissue. They incorporated particles that generated oxygen into the hydrogel through the inverse electron demand Diels–Alder (IEDDA) “click” process to ensure a consistent supply. In situ crosslinking via the “click” reaction enhanced the hydrogel’s mechanical properties and its ability to promote bone tissue healing [190].

The extracellular matrix encases a small aggregation of chondrocytes inside cartilage lacunae. Jeon et al. reproduced this structure in human-derived nasal septal chondrocytes using TA-based clustering and click chemistry-based hydrogel. This technique facilitates efficient patient harvesting, stemness, and differentiation. They used TA binding to create lacunae-like hNC clusters. Hydrogel encapsulation of hNC clusters was achieved using click chemistry linking modified PEG and gelatin. A hydrogel using click chemistry demonstrated increased hNC cluster viability and chondrogenic differentiation. Immunohistochemical staining demonstrated cartilage regeneration after the injection of hNC clusters in hydrogel into a rat osteochondral lesion model. These results underscore the potential of the lacuna-mimic structure for in vivo cartilage regeneration using hNC clusters and click chemistry-based hydrogels [191].

Contemporary bone cement solutions often need free radical or metal-based initiators and catalysts for crosslinking, which may provide risks. The substantial scaffolds have a protracted degradation rate, hence complicating the cellular process of tissue regeneration. Liu et al. created a porous organic–inorganic nanohybrid cement (PO-click-ON) that employs metal-free biorthogonal click chemistry for crosslinking, resulting in a porous architecture akin to natural bone tissue via particle leaching. The strain-promoted click reaction efficiently crosslinks polymer chains without using hazardous initiators or catalysts. PO-click-ON implants exhibited exceptional in vitro adherence of stem cells and facilitated their osteogenic development, with a significant number of stem cells infiltrating the scaffolds extensively. The PO-click-ON device exhibited significant cell adsorption, neovascularization, and osseous development in a rat cranial lesion model. This study introduces a porous click cement those functions as a viable multifunctional platform for bone and tissue engineering [192].

Joint pain and diminished mobility impact 500 million people worldwide owing to the deterioration of articular cartilage (AC) resulting from osteoarthritis (OA). The load-bearing capacity of AC, along with its restricted regenerative potential and particular limitations in patients, has driven research into customized 3D-printed hydrogel scaffolds. Although elaborate multi-network hydrogel scaffolds may be fabricated, a mechanically robust and uncomplicated single-network scaffold is the preferred choice. Arickx et al. investigated the mechanical and swelling characteristics of thiol-ene photopolymerized poly (ethylene oxide) (PEO) hydrogels. Homogeneous polymer networks with diverse topologies were synthesized by selectively crosslinking three-arm (1k) and four-arm (2k) PEO thiol building blocks with linear (6k, 10k, 20k) or four-arm (20k) PEO norbornene.

The assessment of mechanical and swelling properties was performed using nonconfined uniaxial compression testing, supplemented by both empirical and theoretical swelling analysis. Gelation velocity evaluated for single-layer DLP printing. The augmentation of building block branching led to enhanced homogeneity of the hydrogel network and better mechanical and swelling properties. Thiol four-arm PEO (2k) and 20k norbornene hydrogels can withstand 90% compression without failure, exhibiting E-moduli of 895 ± 10 kPa and maximum stresses of 16.4 ± 0.6 MPa, equivalent to the strength of AC. All four-arm thiol gels demonstrated superior printing quality and gelled in 2 s. Hydrogels with enhanced branching demonstrated reduced swelling; yet their mesh size is still unacceptably tiny for biological applications. This research demonstrates that network design results in the formation of resilient and uncomplicated hydrogels [193].

A hydrogel for wound healing was shown by Qu and colleagues. It is composed of chitosan and PF127-aldehydes that were crosslinked via a “click” reaction. This hydrogel has characteristics including adhesion, self-healing capacity, antimicrobial activities, and haemostasis. More than twice as effective in healing as regular commercial dressings [194]. Liu and colleagues developed a hydrogel for osseous healing by combining sodium alginate, gelatin, and polydopamine-decorated nanohydroxyapatite (PHA). The combination was then crosslinked via “click” chemistry [195]. The hydrogel successfully achieved in situ crosslinking at the site of bone injury by “click” chemistry. Experiments indicate that hydrogels enhance bone tissue regeneration, perhaps mitigating postoperative inflammation and addressing the issue of voids after bone injury. Employing “click” chemistry, Ocando and associates crosslinked alginate with alginate/Mg-doped hydroxyapatite (MgHAp) [196]. The synthesized hydrogels served as bio-scaffolds, emulating the porosity of bone tissue and providing a substrate for bone cell adhesion and proliferation, hence facilitating bone tissue regeneration. Recent study indicates that “click” chemistry is essential for the use of hydrogels in tissue regeneration.

### 4.2. Skin and Wound Healing

The skin, being the body’s biggest organ, is essential for safeguarding against several hazards, including diseases, radiation, and severe temperatures. Burns, abrasions, or lacerations are the predominant causes of dermal damage. Although minor skin lesions may heal spontaneously, full thickness burns, and other severe injuries need medical intervention to expedite the healing process. Haemostasis, inflammation, proliferation, and remodeling are the four primary phases in the normal wound healing process. Various cell types, bioactive compounds, and extracellular matrix components are involved in this process. The first phases of wound healing include inflammation and haemostasis. To regulate hemorrhage and eradicate infections, the immune system and platelets are activated throughout both periods [197].

Recent advancements in bioactive modulation demonstrated the role of hydrogels as bioactive materials in activating platelets and regulating the chemotaxis of immune cells, and controlling cell expression [198]. The first recruited immune cells are essential for secreting chemokines and growth factors, which attract other cells and guide the healing process towards the subsequent phase of proliferation. The proliferation phase includes multiple processes, such as the formation of granulation tissue, which entails the creation of a temporary extracellular matrix, vascularization, and re-epithelialization, culminating in the development of an epidermal skin layer and the contraction of the wound surface. This distinctive stage is regulated by a variety of cells, primarily fibroblasts, endothelial cells, macrophages, and keratinocytes. The third step entails the remodeling process, during which the previously produced matrix progressively converts into functioning skin or scar tissue. To accelerate this four-stage healing process, several click chemistry-derived biopolymeric hydrogels, with and without pharmacological agents or growth factors, have been assessed on diverse defect regions [199].

Recent advancements in bioactive modulation demonstrated the role of hydrogels as bioactive materials in activating platelets and regulating the chemotaxis of immune cells, and controlling cell expression [198].

Huang et al. created a hydrogel platform capable of cell printing, using thiolated γ-polyglutamic acid (γ-PGA-SH), glycidyl methacrylate-conjugated γ-polyglutamic acid (γ-PGA-GMA), and thiolated arginine-glycine-aspartate (RGDC) sequences. In this configuration, RGDC functions as an integrin receptor, facilitating cell attachment and dispersion. This platform is created with an optimized gelation technique, emphasizing the efficiency and accuracy intrinsic to the thiol-ene click reaction. The peptide-based hydrogel platform shows potential as effective carriers for cell and growth factor administration, concurrently creating a living material that closely resembles the physicochemical properties of skin tissue [200]. Hu et al. developed a range of novel acid-responsive aminoglycoside hydrogels. Antibacterial hydrogels were synthesized using oxidized polysaccharides, such as dextran, carboxymethyl cellulose, sodium alginate, and chondroitin, in conjunction with crosslinking agents via the Schiff base reaction. The experimental findings demonstrated that the hydrogels had exceptional antibacterial capabilities, both in vitro and in vivo [201].

Lu’s research group focused on wound healing using click chemistry. Employing NIR-responsive substrates, Lu et al. procured cell sheets and affixed fibronectin to their surfaces. They synthesized fibronectin-attached cell sheets (FACS) by metabolic glycoengineering of non-natural sugars, carbon diimide chemistry, and click chemistry. The biological assessment revealed that the binding of fibronectin and the whole biochemical reaction cascade were biocompatible, further promoting cell migration and proliferation. In Vivo experiments on wound healing revealed that FACS efficiently integrated the therapeutic advantages of fibronectin and cell sheets, resulting in superior wound repair compared to conventional cell sheets (CS). FACS have the capability to enhance the regeneration of the dermis, epidermis, hair follicles, and blood vessels. It may accelerate collagen deposition and reduce inflammatory reactions, demonstrating significant practical potential in regenerative medicine [202].

Wang et al. have formulated a multifunctional hydrogel dressing (FHE hydrogel) that is crosslinked via the Schiff base reaction, using Pluronic F127 (F127), oxidized hyaluronic acid (HA), and ε-εpolylysine (EPL). This hydrogel has several activities, including fast haemostasis, self-healing, tissue adhesion, high-efficiency antibacterial activity, and resistance to various drug-resistant bacteria. Exosomes obtained from adipose-derived MSCs were integrated into the hydrogel via electrostatic interactions, yielding a hydrogel that exhibited prolonged pH-responsive release characteristics. Research in rats indicated that exosome-infused hydrogel dressings may augment angiogenesis in injured tissue, therefore promoting wound healing. Furthermore, it may promote cell proliferation, granulation tissue development, collagen deposition, remodeling, and re-epithelialization, leading to expedited healing of severe wounds [203].

Chen et al. developed a self-healing hydrogel-encapsulated chlorhexidine acetate and basic fibroblast growth factor for wound healing. This facilitated the rapid delivery of the chlorhexidine acetate, to inhibit inflammation at the early stages of wound repair, and the sustainable release of the basic fibroblast growth factor to stimulate cell proliferation and wound repair at the wound site. The results indicated that the hydrogel promoted wound healing by reducing inflammation, promoting collagen fiber growth and fibroblast proliferation [204].

### 4.3. Neural and Spinal Cord Scaffolds

The spinal cord is vital to the central nervous system. Sensory and motor impulses are conveyed between the brain and peripheral nerves via the spinal cord. The spinal cord also governs fundamental reflexes, including urination and defecation. Significant central nervous system injuries, including spinal cord injuries (SCI), may impair sensory and motor skills below the lesion site. Individuals with spinal cord injuries often encounter pressure ulcers, urinary issues, breathing difficulties, deep vein thrombosis, pulmonary embolism, spasms, pain, autonomic dysreflexia, and osteoporosis, in addition to neurological deficits. The patient’s quality of life may be diminished, and these complications might obstruct rehabilitation efforts and provide life-threatening dangers. Annually, there are between 13,300 and 22,600 new instances of spinal cord injury globally. Substantial medical costs, prolonged recovery, and employment loss profoundly affect family members. Regeneration of the spinal cord is crucial; however, it poses considerable problems. Hydrogel-assisted tissue engineering may facilitate the healing of spinal cord injury. Stem cells possess the capacity for self-renewal and differentiation into tissue cells, rendering them essential as fundamental cells in tissue regeneration and repair processes. Various cells, including mesenchymal stem cells, neural stem cells, embryonic stem cells, olfactory unsheathing cells, and oligodendrocytes, are transplanted to address spinal cord injury [205,206].

Zhang et al. documented the creation of injectable hydrogels modified with clickable methylprednisolone (MP) and a cellular adhesion peptide, using free radical polymerisation to promote nerve regeneration after spinal cord damage (SCI). Hydrogels conjugated with MP may affect the immunoinflammatory milieu of spinal cord injury and promote neuronal survival. Hydrogels with varying stiffness levels were developed by altering concentration ratios to evaluate appropriate mechanical stimulation. In a dorsal root ganglion model, hydrogels modified with MP and displaying mechanical signals like those in adult rat spinal cords shown improved efficacy in promoting the formation of dorsal root ganglion. Hydrogels modified with MPs has the capability to regulate the immune-inflammatory milieu and improve the restoration of motor and sensory functioning. The positive outcomes demonstrated that the interplay between immunomodulation and mechanical signals is crucial for improving nerve regeneration, underscoring the significant potential of hydrogels as a therapeutic approach for spinal cord injury repair [207].

Li et al. used a crosslinked HA hydrogel combined with a binding PPFLMLLKGSTR peptide to encapsulate MSCs and MnO2 nanoparticles. Animal studies indicated that the implantation of mesenchymal stem cells (MSCs) promoted the development of neuronal cells in spinal cord injury, hence improving motor function recovery in SCI rats [208].

In another study, Tigner et al. evaluated in situ constructed poly (ethylene glycol) (PEG)-based granular hydrogels for the administration of neural progenitor cells (NPC) in a mouse model of spinal cord injury. Thiol-norbornene click chemistry is used to synthesis microgel precursors by the reaction of four-armed PEG-amide-norbornene with peptides that are enzymatically degradable and cell-adhesive. In situ, assembly into scaffolds is accomplished with unreacted norbornene groups in conjunction with a PEG-di-tetrazine linker. The granular hydrogel scaffolds exhibit superior biocompatibility and do not adversely affect the inflammatory response after spinal cord injury. Moreover, the granular hydrogel scaffolds, employed for NPC delivery, enhanced NPC engraftment, did not adversely affect the immune response to the NPC grafts, and successfully encouraged graft differentiation into neuronal or astrocytic lineages, as well as axonal extension into the host tissue. The findings together endorse the use of PEG-based granular hydrogel scaffolds as a suitable biomaterial substrate for NPC distribution and need further research, particularly for more severe spinal cord injuries [209].

In addition to motor function losses, damage at the spinal cord injury site may initiate a series of subsequent pathophysiological reactions. This encompasses ischemia and hypoxia, the invasion of inflammatory cells, and the release of excitatory amino acids and free radicals. These processes may cause the deterioration of axons and myelin sheaths, potentially culminating in the death of nerve cells. Consequently, the incorporation of hydrogel materials at the spinal cord injury site enhances immunomodulation throughout the healing phase of spinal cord injuries. Zhang et al. synthesized an antioxidant hydrogel using the Schiff base reaction, which included crosslinking between aldehyde HA, adipic acid dihydrazide graft HA, and 2,2,6,6-tetramethylpiperidinyloxy. The antioxidant hydrogel exhibited substantial antioxidant properties in the in vitro simulated peroxidized microenvironment. Following implantation, the hydrogel exhibited a significant improvement in motor function recovery in rats subjected to spinal cord transection [210].

Noncovalent interactions between cells and external stimuli are recognized as crucial physiological connections that affect cellular activity. The influence of covalent bonds between cells and biomaterials on cellular behavior has yet to be investigated. Liu et al. presented an approach that combines covalent coupling of biomaterials (collagen fibers/lipid nanoparticles) with several cell types (exogenous neural progenitor cells/astrocytes/endogenous tissue-resident cells) to improve neural regeneration after spinal cord injury. Our results demonstrate that metabolic azido-labeled human brain progenitor cells adhered to dibenzo cyclooctyne-modified collagen fibers significantly improved cell adherence, spreading, and differentiation relative to noncovalent adhesion. Dibenzo cyclooctyne-modified lipid nanoparticles encapsulating edaravone, a recognized reactive oxygen species scavenger, may also selectively target azide-labeled spinal cord tissues or transplanted azide-modified astrocytes to improve the spinal cord injury microenvironment. In a rat spinal cord injury model, the use of several covalent conjugation techniques collectively improved neural regeneration, suggesting that the interactions between cells and biomaterials are very promising for tissue regeneration [211].

### 4.4. Ocular and Corneal Regeneration

The cornea is a transparent, thin tissue that envelops the front portion of the eyeball. It constitutes one-sixth of the thickness of the extraocular fiber membrane, functioning to mechanically safeguard the eyeball and prevent the ingress of dangerous chemicals into the eye. The cornea operates akin to a camera lens, allowing light to traverse the pupil and concentrate on the retina’s funds for imaging. The cornea, being the outermost layer of ocular tissue, is very vulnerable to damage. Corneal injury may result from ocular trauma, including mechanical and chemical traumas, in addition to bacterial infections, burns, and viral infections. The repair of corneal injuries involves several components, including growth factors, extracellular matrix remodeling, and the participation of corneal cells, namely epithelial, stromal, and endothelial cells. Thus, the regeneration of the cornea poses considerable difficulties. Researchers are developing novel materials for corneal restoration and regeneration as tissue engineering advances in regenerative medicine. Materials for corneal regeneration must closely resemble the normal cornea regarding water content, light transmittance, and ion transmission rate. They must provide enough mechanical strength to withstand surgical sutures and post-operative care, while simultaneously exhibiting stability (resistance to enzymatic hydrolysis) and biocompatibility to promote the proliferation of corneal epithelial cells. Research on corneal regeneration mostly focusses on hydrogels because of their critical properties [212,213].

Hydrogels based on poly (ethylene glycol) (PEG) are widely acknowledged as highly promising three-dimensional scaffolds for cells, finding extensive applications in tissue regrowth and regeneration. Nonetheless, obtaining functionalized PEG hydrogels that offer dynamic, cell-instructive microenvironments continues to be inherently difficult. Lei et al. utilized the specificity of the click reaction to develop a range of hydrogels made from 4-arm PEG tetraazide (4-arm-PEG-N3) and di-propagated peptides (GRGDG and GRDGG), showcasing customizable physicochemical properties ideal for 3D cell scaffolds. The interaction between the azide groups of PEGS and the alkynyl groups of peptides, facilitated by copper, resulted in the formation of triazole rings, leading to the creation of a crosslinked hydrogel. The duration of gelation and the mechanical integrity of the hydrogels varied according to the PEG/peptide feed ratio. The hydrogel exhibited a distinctive porous structure along with suitable swelling characteristics. The cytotoxicity assay conducted in vitro showed that the resulting hydrogels displayed no notable cytotoxic effects on human corneal epithelial cells (HCECs). The co-incubation with HCECs had a significant impact on cell adhesion, spread ability, and proliferation, which depended on the density of RGD, and peptide sequences present in the hydrogels. The hydrogel proposed showed notable ocular biocompatibility following subconjunctival implantation in the eyes of rabbits. The findings suggest that biofunctional hydrogels made from PEG and RGD motifs, created through a controlled click reaction, could be effective as 3D cell scaffolds for the regeneration of corneal epithelium [214].

Li et al. created photocurable hydrogels using acrylate and thiolated gelatin as a reparative substance intended for the regeneration of localized corneal injuries. Animal experiments demonstrated that the hydrogel had remarkable biocompatibility in vivo with rabbit corneas and facilitated the healing of localized corneal injuries, attaining epithelial wound covering after just 3 days [215].

Rosenquinst et al. created an injectable hydrogel in a separate investigation by crosslinking acetyl thiol collagen with PEG-maleimide via a thiol-maleimide click reaction. Upon combining the gel constituents, this hydrogel may be positioned in an aqueous buffer while preserving its form. The synthesized hydrogel can be maintained in an aqueous buffer for at least one year without significant swelling or changes in form. The mechanical characteristics of hydrogels may be readily modified by changing the ratio of the two components, obviating the need for new synthesis or design. The created thiol collagen hydrogels are entirely transparent, enzymatically degradable, and promote the adhesion and proliferation of human corneal epithelial cells. The hydrogels produced demonstrate a lap shear strength akin to that of fibrin glue, regarded as the gold standard for tissue adhesives in clinical applications, and possess the ability to seal porcine corneas, allowing them to withstand a higher mean pressure than the average intraocular pressure in humans. This discovery lays the groundwork for the future use of this hydrogel in healing corneal perforations, in situ tectonic filling of corneal holes, and the possible delivery of therapeutic cells and proteins [216].

Koivusalo et al. developed tissue sticky HA-DOPA hydrogels by integrating dopamine moieties into hydrazine crosslinked hyaluronic acid for corneal regeneration. The entrapment of human adipose-derived stem cells (hASCs) in HA-DOPA hydrogels exhibited advantageous proliferation and cellular elongation properties. The HA-DOPA hydrogels exhibited superior tissue adherence after implantation in a porcine ocular organ. This hydrogel has significant promise as a reparative material for facilitating corneal regeneration, as seen by the findings [217].

### 4.5. Bioprinting and 3D Fabrication Using Click-Based Inks

Recently, diverse reaction techniques have facilitated the creation of new innovative multifunctional hydrogel biomaterials, identified as bioinks for 3D bioprinting [218]. The hydrogels provide a crucial milieu by replicating the extracellular matrices present in natural tissue structures. Diverse hydrogels, both natural and manufactured, demonstrate multifunctional attributes including biocompatibility, elevated mechanical strength, pH responsiveness, and thermal sensitivity. These hydrogels, when integrated with nanomaterials, composite materials, growth factors, and biomolecules, have been used to create complex 3D structures and functional tissue constructions using 3D bioprinting technology for tissue engineering applications [219]. A multidisciplinary approach integrating stimuli-responsive biomaterials, click chemistry, and advanced 3D bioprinting technology may improve the feasibility of producing complex 3D structures that are functional and capable of resembling native tissues in tissue or organ formation [220,221].

Bock et al. encapsulated patient-derived breast cancer cells in bioprinted hydrogels composed of polyethylene glycol (PEG), which were functionalized with adhesion peptides such as RGD, GFOGER, and DYIGSR, alongside gelatin-derived hydrogels including gelatin methacryloyl (GelMA) and thiolated-gelatin crosslinked with PEG-4MAL (GelSH) [222]. Li et al. demonstrated click chemical processes using CuAAC to synthesize peptide-functionalized poly (ester urea) scaffolds intended for bone tissue engineering applications. L-phenylalanine functionalized with propargyl groups was used in poly (ester urea) scaffolds in conjunction with BMP-2 and osteogenic growth peptide. The scaffolds exhibited superior osteogenic activity and hMSC differentiation relative to the nonfunctionalized scaffolds [223].

Based on recent research, High sensitivity in molecular identification inside biological contexts may be attained by surface-enhanced Raman spectroscopy (SERS). Incorporating SERS sensors into 3D models may provide resilient platforms for investigating biological responses to medicines, metabolic pathways, signaling, and intercellular interactions. Afonso et al. introduced a library of plasmonic hydrogels including gelatin, alginate, and carboxymethylcellulose, which may be orthogonally photo-crosslinked using thiol–ene click chemistry. They also determined the main physicochemical parameters that enhance their SERS performance. Swelling, porosity, and chemical composition were recognized as critical elements affecting their capacity to detect diverse compounds by SERS. Furthermore, the biocompatibility and printability of these hydrogels must be evaluated to ensure they meet the standards for use as 3D cellular scaffolds, emphasizing their capability for real-time and in situ detection of biorelevant metabolites [94].

In another study, Bebiano et al. proposes a bioinspired design methodology for bioengineering double crosslinked pectin-based bioinks. This technique seeks to emulate the mechanical characteristics and organization of cell-adhesive ligands and protease-sensitive regions present in the dermal extracellular matrix, thereby enabling the bioprinting of bilayer 3D skin models. Pectin modified with methacrylate acted as a primary biomaterial, enabling hydrogel production by either chain-growth or step-growth photopolymerization. This method facilitated the autonomous modulation of bioink rheology, in conjunction with the mechanical and biochemical cues present in the cellular environment. By modifying the concentrations of the crosslinker and polymer in the bioink formulation, dermal constructions were bioprinted with varying stiffnesses that were physiologically relevant, resulting in significant site-specific differences in the morphology and spreading of dermal fibroblasts. The synthesized thiol-ene photo-clickable bioinks facilitate the bioprinting of skin models in diverse configurations, aiding in the reconstruction of both dermal and epidermal layers. The tailored bioinks expand the range of printable biomaterials for the extrusion bioprinting of 3D cell-laden hydrogels and provide a versatile platform to investigate the influence of material cues on cell destiny, offering prospects for in vitro skin modeling [224].

Stichler et al. recently published a comparable method that integrates thiol-ene click chemistry with 3D bioprinting to fabricate three-dimensional objects. Polyglycidol-based hydrogels were used, crosslinked by UV light, and cytotoxicity assessments were performed using hMSCs derived from bone marrow. High molecular weight hyaluronic acid was integrated to alter the rheological characteristics, enabling the printing of 20-layered structures with remarkable repeatability. Recently, several scientists have documented this combinatorial method that employs click chemistry in conjunction with 3D bioprinting for diverse applications in tissue engineering [225,226].

## 5. Advantages and Limitations of Click Chemistry in Hydrogels

Click chemistry provides rapid, selective, and bioorthogonal hydrogel formation with tunable mechanical and biological properties, while enabling cell and bioactive molecule incorporation. Metal-free and catalyst-free strategies further improve biocompatibility. Key advantages and limitations are summarized in Figure 5 and discussed in detail in the following subsections. Limitations include potential cytotoxicity, heterogeneous crosslinking, challenges in controlling stability and degradability, and the cost of specialized precursors.

### 5.1. Advantages

**Rapid Gelation under physiological conditions:** The most prominent advantage of click chemistry is the ability to synthesize hydrogels very quickly under physiological conditions. Crosslinking of the structure can take as little as a few seconds to as long as a few minutes. Meanwhile all of these can happen in the presence of living cells and bioactive molecules so that makes it minimally invasive in situ curing of implants. For example, thiol–ene chitosan gels can crosslink in under 30 s for drug delivery purposes, while SPAAC-based PEG/elastin-like polypeptide hydrogels for neural tissue formation achieve gelation in approximately 20 min without requiring any cytotoxic catalysts. This rapid gelation can be useful in preventing precursor leakage. Another important effect of rapid gelation is the ability to use high speed fabrication processes throughout the process, such as 3D printing, which require high crosslinking speeds [227].

**Bio-orthogonality and Cytocompatibility:** Click reactions are biorthogonal as discussed in earlier sections meaning they are performed without interfering with biological molecules such as proteins, nucleic acids or cell membranes. SPAAC and inverse demand Diels–Alder (IEDDA) hydrogels are proven to be more than 95 percent cell viable in the encapsulation studies This cell viability occurred even when the hydrogel crosslinked directly in the presence of cells. Since the click chemistry reactions generate minimal or even nontoxic byproducts and avoid harsh PH numbers and use mild aqueous solvents; it is useful for embedding additives like growth factors, antibodies and extracellular matrix proteins directly into the hydrogel during formation [89].

**Catalyst Free and Metal Free Strategies:** The traditional click chemistry method (CuAAC) is highly efficient, but it lacks biocompatibility due to copper induced cytotoxicity. This led to development of newer and less toxic methods like SPAAC, thiol-ene and certain DA systems. These methods eliminate the need for toxic metal catalyst hence improving the biocompatibility significantly. It is worth noting that using copper free method not only improves this property but also improves the fabrication process by simplifying the post processing since extensive copper removal steps such as dialysis or chelation are unnecessary, reducing time and cost and residual catalyst contaminations [228].

**Tunable Mechanical and Chemical properties:** In the click hydrogels there is fine control on stiffness, elasticity and degradation rates by adjusting the crosslinker density, side chains and linkages chemistry. This tunable property can enhance the mechanical properties so that it matches the native tissue’s property. One more advantage is that the degradation can be controlled by introducing hydrolytically or enzymatic linkers that allow scaffolds to gradually transfer load so the regeneration would be faster and more efficient [229].

Gustavo et al. investigated the effect of molecular weight and size on PEG-based, metal-free catalyzed hydrogels, demonstrating their influence on mechanical properties and gelation time. Increasing the molecular weight enhanced the elasticity of the hydrogels. While these hydrogels were designed to be porous for skin regeneration, controlling the molecular size also allowed for improved mechanical performance [230].

Collins et al. used the increasing of the degree of crosslinking strategy for improving their oxime-boronate bonded hydrogel. The use of boronate linkage increased the crosslinking so that the doubled the shear modulus of the hydrogel [231].

Shao et al. investigates the dynamic and static mechanical properties of the CNC-PEG click based hydrogel. The resultant hydrogel showed fracture elongation of about 700%. Also, the fatigue related properties are investigated for dynamic situation performance showed self-recovery nature of the bonds used helped for antifatigue properties of the hydrogel. This study highlights not only the Mw and crosslinking density are important factor on the mechanical properties but also the bond’s nature in the molecular structure plays a crucial role especially in the dynamic situations like fatigue [232]. The stress relaxation of the hydrogel via using reversible crosslinking is what that researched by Tan et al. by varying the ratio of 4-armPEG-azide to 4-armPEG-amide-DBCO (ranges from 1:0.6 to 0.6:1) using SPAAC crosslinking they achieved various mechanical properties. Perfectly crosslinked hydrogels showed balance between stiffness and elongation while excess azide gels showed a decrease in compressive strength and increased swelling ratio due to fewer covalent crosslink linkages [233]. In another study the tunable mechanical properties of the boronate-based hydrogels achieved using click chemistry is investigated. The use of SPAAC method resulted in mechanical stability under hMSC culture condition by preventing the gel from erosion in media. The other result of various crosslinking density is not only the moduli will change but also the relation behavior of gel also changed. The mechanical robustness of the SPAAC crosslinked hydrogels in the study highlights dynamic matrix remodeling, making them ideal for studying mechanotransduction especially in viscoelastic environments [234].

Wei et al. studied a dextran-based self-healing hydrogel fabricated using reversible Diels–Alder click chemistry. Using fulvene modified dextran as the main polymer chain. The theological examination showed the structural recovery that confirms shear thining behavior. Complete healing of hydrogel showed to occur after only 7 h in physiological temperature [235].

Tan et al. also investigated the self-healing characteristic of a hyaluronic-based hydrogel formed using reversible Diels–Alder click chemistry. This gel showed transition from gel-like structure to solid form after about 25 min. Although it is relatively high time compared to other typical methods of click chemistry, the resultant hydrogel showed great potential balancing specificity in proteins encapsulation, cytotoxicity and stable mechanical properties after 20 days [233]. The multiple chemical modification is another way to introduce same hydrogel platform with, for example, two reactive systems such as aldehyde-azide hydrogels. These can undergo both Huisgen cycloaddition and Schiff-base reactions which can be enabling simultaneous presentation of bioactive ligands, dyes or drug molecules.

Yu et al. used combination of Diels Alder click reaction and acylhydrazone bond. The DA reaction maintained the hydrogel’s mechanical properties and makes the basis of structural integrity of hydrogel while the dynamic covalent acylhydrazone bond resulted in self-healing property of hydrogel [125].

Liu et al. investigated Diels–Alder click reactions as a tool for tuning the mechanical properties in hydrogels. The kinetic control via DA click reaction enabled precise controlling of crosslinking density and rate, which influenced stiffness, viscoelasticity and self-healing capabilities of hydrogel. The other thing studied in this study was faster kinetic reaction; Electron-demand, click reaction showed denser molecular structure which enhanced the mechanical toughness of the hydrogel [236].

Another advantage of this controllability is the ability to design hydrogels that can be triggered or patterned using light or ultrasound. For example, visible light has been used to activate thiol–ene hydrogels for bone fragment adhesion and regeneration, while focused ultrasound has been applied to trigger Diels–Alder PEG hydrogels for localized release of bone morphogenetic protein-2 (BMP-2) [134].

### 5.2. Disadvantages

**Biocompatibility Concerns from Catalysts, Light and Reactive Intermediates:** Although CuAAC has remained one of the most efficient click reactions but the copper catalyst even at trace levels is highly toxic. This toxicity comes from the tendency of copper to leave harmful residues capable of causing oxidative stress. Also, the other issue with copper is strong binding of copper to the biomolecules that makes the attempts to remove the copper complicate. The photochemical click hydrogels use light as initiator, but this also may cause damage to DNA, denaturing proteins and releasing cytotoxic radicals even in metal free ones. One of the notable works for reducing the toxicity effect of copper was performed by Del Amo et al. in their work the biocompatibility of CuAAC click is improved by using a tris(triazolylmethyl)amine-based ligand for Cu(I). The new copper-based catalyst allowed noninvasive usage for imaging usage of early embryogenesis of zebrafish [237]. The copper biocompatibility improvement is subject of study in Li and Zhang study. The method includes development of copper-chelating ligands and chelating azide reagents to improve both biocompatibility and reaction efficiency. The work successfully used in labeling specific biomolecules [238].

**Mechanical Limitations:** Despite their chemical efficiency, click hydrogels lack the proper toughness of conventional hydrogels. This low toughness will make hydrogel less energy absorbent hence making it less useful where mechanical properties are an issue. The very rapid reaction kinetics of hydrogels in click chemistry make it useful in synthesizing process but causing incomplete mixing of precursors leading to heterogeneous crosslinking distribution, weak spots and result in poor mechanical properties.

**Challenges in Degradation Control:** In some click reactions the click linkages are designed so to be stable for long term use in the body. The main feature of these linkages is near-complete resistance to hydrolysis that makes them stable for long periods in physiological condition. While this property is advantageous in some conditions but can lead to hindering the tissue remodeling and delay scaffold clearance from the body. On the other hand, more dynamic bonds like hydrazones or oximes may degrade too rapid under physiological conditions like PH. Achieving the optimal balance between stability and degradability remains a major challenge with many systems degrading too quickly yielding acidic byproducts or too slowly persisting longer than desired [239,240].

**Complex and Costly Manufacturing:** Many click hydrogel systems rely on polymers that have been pre-functionalized with azide, alkyne, thiol, furan, norbornene, or tetrazine groups. These chemical modifications often come with price. The price is not only in terms of precursors price but also the are problems like multiple synthetic steps, expensive reagents and low yield processes. Cyclooctyne crosslinkers used in SPAAC can be labor intensive to synthesize and costly at scale. Very high kinetics make them challenging to achieve homogenous mixing in large volumes or during extrusion 3D printings. One other challenge is to provide high intensity and uniform light sources which can be technically demanding. Farahani et al. used near UV light exposure for oxime ligation, the reaction between alkoxyamines and aldehydes has been performed. This photochemically controlled click chemistry strategy was beneficial for tunable properties but also addressed the technical challenges involved with this method [53].

## 6. Summary and Outlook

Click chemistry-based hydrogels are emerging as a versatile and powerful platform for regenerative medicine, tissue engineering, and drug delivery. Their precision, modularity, and unique adaptability position them to overcome many of the limitations associated with traditional biomaterials, such as uncontrolled crosslinking, poor reproducibility, and limited biological integration. By providing powerful yet tunable chemical strategies, click reactions allow the design of hydrogels with highly defined structures, predictable functionality, and flexibility to incorporate diverse bioactive cues.

Importantly, these systems are not limited to academic development. They have great potential for real world applications. Their compatibility with aqueous environments, rapid reaction kinetics, and their orthogonality to biological systems make them particularly suitable for clinical applications. As a result, click-based hydrogels may accelerate the development of advanced wound dressings, injectable therapies, ophthalmic adhesives, and scaffolds for complex tissue regeneration. However, realizing this promise will require ongoing interdisciplinary collaboration. Chemists must continue to refine reaction tools for safety and efficacy. Engineers must advance scalable manufacturing and delivery methods; biologists must evaluate cellular responses and regenerative outcomes; and physicians will play a pivotal role in defining practical needs and guiding regulatory transformation. Ultimately, the convergence of chemistry, biology, and engineering in this field may reshape the future of biomedical sciences and translate into tangible healthcare solutions that improve quality of life.

## Figures and Tables

**Figure 1 gels-11-00724-f001:**
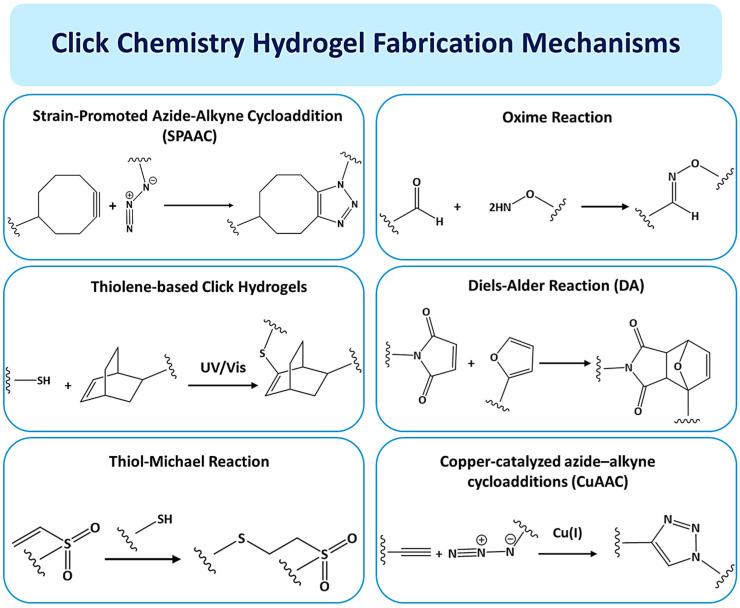
Schematic of the most important click chemistry reactions based on their various mechanisms.

**Figure 2 gels-11-00724-f002:**
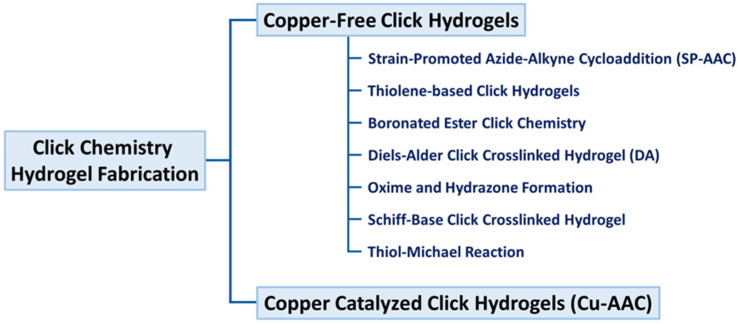
Types of hydrogel fabrication mechanisms based on the click reaction chemistry.

**Figure 3 gels-11-00724-f003:**
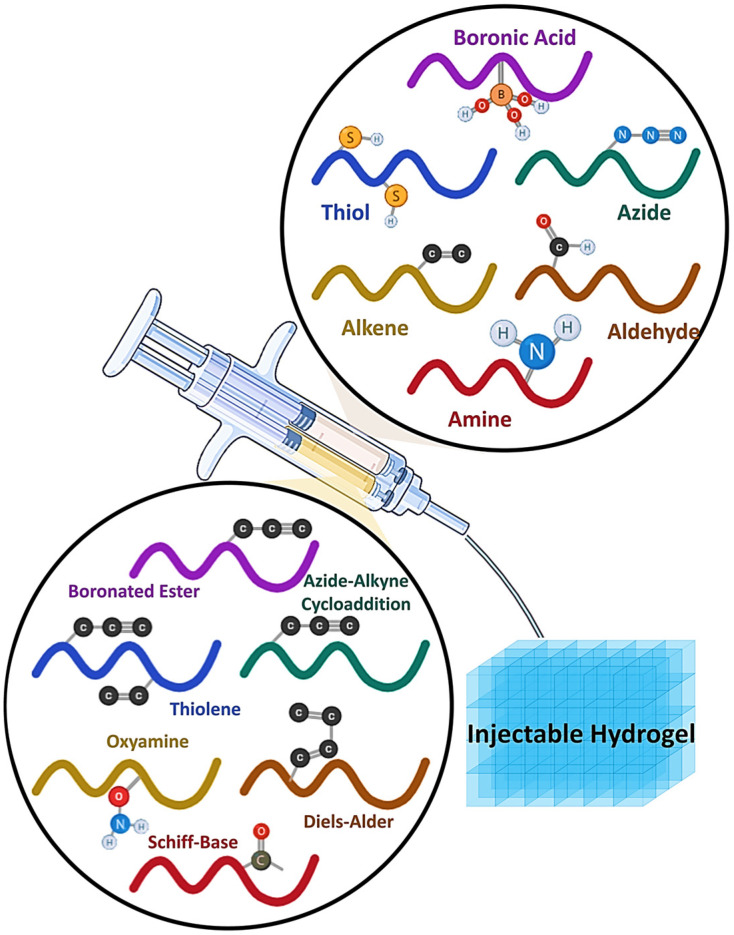
Illustration of functional groups and dynamic covalent chemistries employed to design injectable click chemistry hydrogels, where reagents of matching colors undergo specific reactions under defined conditions to form crosslinked networks with tunable properties, self-healing ability, and excellent biocompatibility for biomedical applications.

**Figure 4 gels-11-00724-f004:**
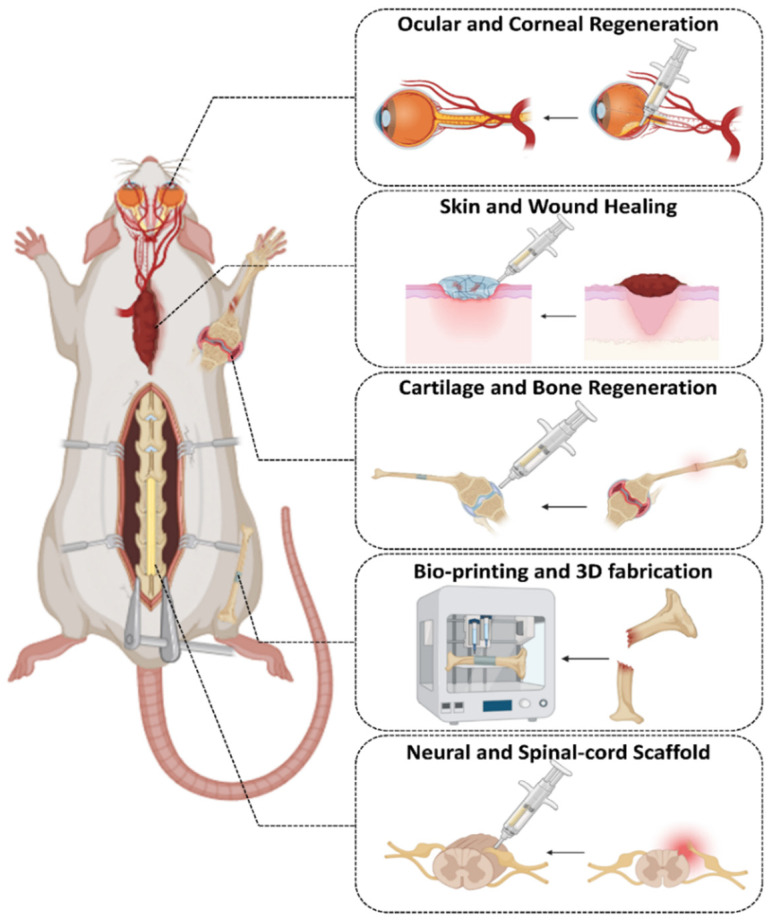
The multidisciplinary pillars of modern regenerative medicine: showcasing advanced research in skin wound healing, corneal and ocular repair, cartilage and bone regeneration, neural scaffold engineering for spinal cord injuries, and the transformative technology of 3D bioprinting and fabrication.

**Figure 5 gels-11-00724-f005:**
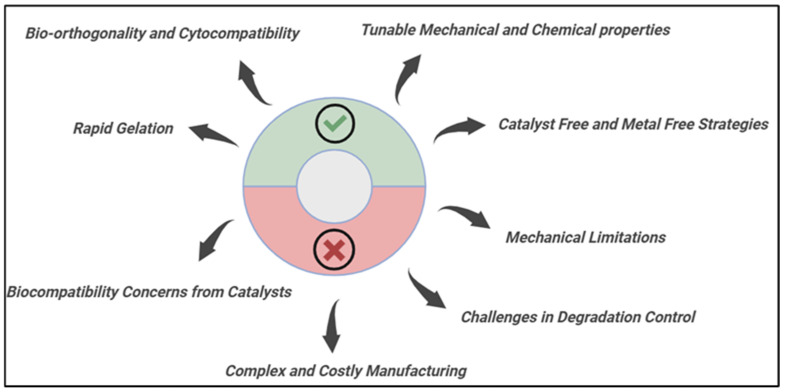
Summary of advantages and disadvantages of click chemistry synthesis methods.

**Table 1 gels-11-00724-t001:** Common click chemistry reactions used for hydrogel crosslinking in tissue engineering applications.

Reaction Type	Mechanism	Advantages	Limitations	Applications	[Ref.]
**Azide–Alkyne Cycloaddition**	Copper(I)-catalyzed or strain-promoted reaction between azides and alkynes	High specificity, bioorthogonal, efficient under mild conditions	Copper catalyst may be cytotoxic (CuAAC); strain-promoted versions more expensive	Injectable hydrogels, cell encapsulation, drug delivery	[32,33]
**Thiol–Ene Reaction**	Radical-mediated addition of thiols to alkenes	Fast kinetics, no metal catalysts, tunable crosslinking	Requires photoinitiator or thermal initiator	In situ gelation, biofunctionalization, 3D cell culture	[34,35]
**Diels–Alder Reaction**	[4+2] Cycloaddition between a diene and a dienophile	Reversible under thermal control, catalyst-free	Temperature sensitivity may affect biological components	Injectable hydrogels, drug release systems	[16,36]
**Michael Addition**	Nucleophilic addition of thiols to electron-deficient vinyl groups	Mild conditions, high yield, no need for catalyst	Sensitivity to pH and competing nucleophiles	Cell-laden hydrogels, tissue engineering scaffolds	[37,38]
**Oxime/Hydrazone Formation**	Reaction between aldehydes/ketones and hydrazides or aminooxy groups	Chemoselective, compatible with aqueous environments	Hydrolysis-sensitive; slow gelation in some cases	Injectable matrices, bioadhesives	[39,40]
**Tetrazine–Norbornene Reaction**	Inverse electron-demand Diels–Alder reaction	Ultra-fast kinetics, catalyst-free, excellent bioorthogonality	Tetrazine compounds can be costly	Real-time cell encapsulation, dynamic biomaterials	[41,42]
**SuFEx (Sulfur Fluoride Exchange)**	Click reaction involving sulfur-fluoride bonds	High yield, stable products, emerging tool in bio-click chemistry	Limited current use in hydrogels; less biocompatibility data	Future directions in robust hydrogel crosslinking	[43,44]

**Table 2 gels-11-00724-t002:** Recent research on azide–alkyne cycloaddition-based hydrogels for tissue engineering, including both (CuAAC) and (SPAAC) approaches.

Hydrogel Composition	Click Type	Application	Ref.
Alginate grafted with BCN/cyclooctyne; azide-peptides (e.g., Az-cRGD)	SPAAC	Peptide functionalisation of alginate preserving peptide bioactivity for 2D/3D cell assays; cell-friendly, in-presence-of-cells coupling.	[74]
Cyclooctyne-modified alginate (ALG-K) conjugated to bis-azide MMP-sensitive peptides (PVGLIG)	SPAAC	Tunable protease-sensitive alginate hydrogels that support cell spreading and matrix remodeling; modular and bioorthogonal.	[75]
Review covering many metal-free click reactions (SPAAC emphasis) applied to PEG, HA, alginate, gelatin hydrogels.	SPAAC & other metal-free clicks	Comprehensive summary of metal-free click chemistries for cell-laden hydrogels and translational potential; practical reagent notes.	[76]
Review/analysis of kinetics across click classes (CuAAC, SPAAC, IEDDA, etc.) relevant to hydrogel design.	CuAAC vs. SPAAC (comparison)	Practical guidance: CuAAC = fast but requires Cu(I) management; SPAAC = biocompatible (no copper) but reagent cost/kinetics tradeoffs—crucial when choosing for cell-laden gels.	[77]
Alginate hydrogels microstructured and functionalised via SPAAC for injectable cell delivery and morphogenesis guidance.	SPAAC	Demonstrated injectable, microstructured, cell-instructive hydrogels produced using bioorthogonal SPAAC; supports 3D cell patterning.	[78]
Alginate–gelatin hybrid hydrogels dual-crosslinked using SPAAC (bioorthogonal) plus ionic or secondary covalent crosslinks.	SPAAC (primary) + secondary crosslinks	Improved mechanics and cell-instructive environment; on viscoelastic tuning for cardiac TE (long-term in vivo results still pending)	[79]

**Table 3 gels-11-00724-t003:** Recent studies on thiol–ene click hydrogels for tissue engineering, covering both synthetic and natural polymer systems.

Hydrogel Composition	Application/Highlight	Ref.
Methoxy-PEG grafted to keratin via thiol-ene photoclick	Eco-friendly fabrication of keratin-based hydrogel, potential for biomaterial scaffolding.	[104]
Thiol-ene hydrogel combined with poly (propylene fumarate) (PPF) scaffold	Local PTH delivery in critical-size bone defects, sustained bioactivity through 21 days, promoting bone healing.	[105]
Gelatin–hyaluronic acid modified for thiol-ene crosslinking	3D-printable cryogel scaffolds, enhanced mechanical stability, biocompatibility, potentially useful in cartilage or soft tissue.	[106]
Hydroxypropyl methylcellulose (HPMC) derivatives via thiol-ene, forming PNP hydrogels with PEG–PLA nanoparticles	Injectable polymer–nanoparticle hydrogels featuring tunable mechanics, retention time, and bioactive peptide loading, ideal for cell/drug delivery.	[95]
Alginate/gelatin/CMC functionalized with thiol/norbornene for thiol-ene photo-crosslinking, incorporating SERS nanorods	3D-printed plasmonic scaffolds enabling real-time metabolic sensing in 3D cell cultures, biosensing + structural scaffold in one.	[94]

**Table 4 gels-11-00724-t004:** Representative studies on boronate ester–crosslinked hydrogels for tissue engineering.

Hydrogel Composition	Crosslink Type (Boronate-Based)	Application/Highlight	Ref.
Cellulose derivatives functionalized with boronic-acid/diol motifs to form reversible boronate esters.	Boronate-ester dynamic crosslinking (self-healing)	Self-healing, conductive hydrogel with good cytocompatibility, candidate for soft tissue repair and bioelectronic interfaces.	[117]
Various polymer backbones (PVA, PEG, polysaccharides) crosslinked by bis-boronic acids to give glucose-responsive networks.	Boronate ester crosslinks responsive to glucose/pH	Closed-loop insulin delivery platforms illustrating how BE hydrogels can be tuned for biomedical, implantable drug-release applications.	[118]
Thin-film hydrogels of polymers bearing phenylboronic acid (PBA) for glucose binding and reversible crosslinking.	PBA–diol boronate esters (glucose competitive)	Glucose-sensitive swelling/optical response; useful as biosensing layers and for smart biomaterials in regenerative medicine workflows.	[119]
Hyaluronic acid (HA) chemically modified with phenylboronic acid groups to form reversible networks with diol-bearing partners.	PBA–diol boronate ester network (dynamic)	Supports 2D/3D culture of articular chondrocytes, promising for cartilage tissue engineering and cell-friendly, dynamic ECM mimicry.	[120]
Gelatin modified with phenylboronic acid crosslinked to oxidized dextran (diol/aldehyde components present) to form an injectable gel.	Boronate ester (PBA–diol) dynamic crosslinks (plus secondary interactions)	Injectable, biodegradable hydrogel for bone TE; good cell viability and in vitro osteogenic markers reported.	[121]
Multi-component hydrogel combining PBA-functionalized polymers with catechol/diol units and ROS-scavenging moieties.	Boronate ester dynamic bonds (pH/glucose/ROS responsive)	Injectable, multi-stimuli-responsive system demonstrating controllable drug release, antioxidant effect and promising in vitro biocompatibility for wound/tissue repair.	[122]
Review summarizing PEG, PVA, HA, dextran, gelatin systems that exploit BE dynamics at physiological pH.	Survey of boronate ester chemistries, pH tuning, and strategies to operate near physiological conditions	State-of-the-art review covering materials design, pH/glucose responsiveness, self-healing, and translational challenges (stability, in vivo data, scale-up). Good starting point for application-directed design.	[123]

**Table 5 gels-11-00724-t005:** Representative studies on oxime and hydrazone–crosslinked hydrogels for tissue engineering.

Hydrogel Composition	Crosslink Type	Application/Highlight	Ref.
Aldehyde-PEG + aminooxy-PEG (oxime), Rapid in situ gelation on wet tissue; physiological pH	Oxime	Strong wet-tissue adhesion/retention; anti-fouling surface	[153]
HA-aldehyde + aminooxy crosslinker, Physiological pH/ionic strength	Oxime	Transparent, vitreous-like modulus; enzymatic stability	[147]
HA-aldehyde/HA-hydrazide + benzaldehyde/hydrazide, Mild, aqueous; tunable stress relaxation	Hydrazone (acylhydrazone)	Viscoelastic (stress-relaxing) networks modulate MSC spreading & secretome	[154]
HA-aldehyde + dihydrazide crosslinker, Fast gelation; pH-responsive	Hydrazone (acylhydrazone)	Self-healing, enhanced mechanics; cytocompatible	[155]

**Table 6 gels-11-00724-t006:** Representative studies on Schiff-base–crosslinked hydrogels for tissue engineering.

Hydrogel Composition	Crosslink Type	Application/Highlight	Ref.
Alginate dialdehyde (ADA) + gelatin (GEL)	Schiff base (imine) between ADA aldehydes and GEL amines; printable grids	3D-printed ADA–GEL scaffolds for cartilage tissue engineering; supports chondrocyte function.	[178]
Oxidized alginate + chitosan (plus conductive phase)	Schiff base (aldehyde–amine) gives self-healing; printable under mild conditions	Conductive, self-healing, 3D-printable hydrogels aimed at TE scaffolds (soft tissues).	[179]
Hyaluronic acid + oxidized chitosan (OCS)	Schiff base; physiological, tissue-adhesive, self-healing	Antibacterial, adhesive injectable hydrogel; promising wound-healing/soft-tissue TE dressing.	[180]
Oxidized dextran (ODex) + gelatin (GEL) + nano-fillers	Schiff base; injectable, self-healing	Growth-factor-free bone regeneration in vivo; enhanced osteogenesis from dynamic imine network.	[185]
Oxidized alginate + gelatin (with microspheres)	Schiff base; in situ gelation	Biocompatible composite scaffolds with improved mechanics for TE applications.	[185]
Gelatin + hyaluronic acid (one oxidized)	Dynamic mine; self-healing	Biodegradable, conductive, self-healing hydrogel; broadly relevant to TE and wound repair.	[186]
Quaternary-ammonium chitosan + oxidized partner	Schiff base; mild, injectable	Adhesive, antibacterial hydrogels suitable for tissue repair dressings and cell delivery.	[187]
Gelatin + oxidized alginate (Schiff base) + NPs	Imine crosslinking; injectable	Localized therapy platform: ADA–GEL imine network is a canonical TE matrix (drug/cell co-delivery).	[188]

## Data Availability

Not applicable.

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
