# Peer review of "Click Chemistry-Based Hydrogels for Tissue Engineering"

_gels, 2025, doi:10.3390/gels11090724_

Round 1
Reviewer 1 Report
Comments and Suggestions for Authors
Review Report
The manuscript titled “Click Chemistry-Based Hydrogels for Tissue Engineering” provides a well-organized and insightful overview of click chemistry strategies for hydrogel development in tissue engineering. It effectively highlights the advantages of click reactions, including CuAAC, SPAAC, thiol-ene/yne, Diels–Alder, and tetrazine–norbornene systems, emphasizing their roles in achieving multifunctionality, injectability, biodegradability, and responsiveness.
Authors can find the comments and questions below.
The introduction should better explain terms such as orthogonality and pseudo-click reactions with practical biomedical relevance. Furthermore, more recent references should be incorporated across all sections to maintain novelty and strengthen the scientific foundation of the article. Below are specific concerns:
Concerns:
- Lines 137–142: The discussion on copper-free click chemistry should expand on the significance of pseudo-click reactions. Add current examples of hydrogel applications Suggest citing, A motion-responsive injectable lubricative hydrogel for efficient Achilles tendon adhesion prevention;doi: https://doi.org/10.1016/j.mtbio.2025.101458 and doi: https://doi.org/10.1016/j.bioactmat.2025.06.004.
- Lines 565–569: The advantages of oxime click chemistry over thiol-based linkages need deeper explanation. Support with examples from tissue engineering and sensor-responsive hydrogels. Consider referencing, Electrospun green fluorescent-highly anisotropic conductive Janus-type nanoribbon hydrogel array film for multiple stimulus response sensors; doi: https://doi.org/10.1016/j.compositesb.2024.111933 & doi: 10.1038/s41467-024-51734-7
- Lines 923–927 – Strengthen the wound healing section by integrating recent findings on hydrogel–immune cell interactions and bioactive modulation of the healing cascade. Suggest citing,
- Lines 1162–1170: The bioink design discussion should include recent developments in multifunctional and bioinspired hydrogel bioprinting. Suggest citing, (a) doi: https://doi.org/10.1016/j.seppur.2024.130597(b)doi:https://doi.org/10.1016/j.cis.2025.103470
- Ensure consistent abbreviation usage (e.g., CuAAC, SPAAC).
- Revise figure captions to be more descriptive, highlighting mechanisms or applications.
- Improve readability by reducing redundancy and polishing sentence structure.
Major revisions are necessary to enhance scientific depth, address specific line-level concerns, and incorporate the most current literature.
Author Response
Comment 1:
Lines 137–142: The discussion on copper-free click chemistry should expand on the significance of pseudo-click reactions. Add current examples of hydrogel applications Suggest citing, A motion-responsive injectable lubricative hydrogel for efficient Achilles tendon adhesion prevention; https://doi.org/10.1016/j.mtbio.2025.101458 and doi: https://doi.org/10.1016/j.bioactmat.2025.06.004.
Response 1:
We appreciate your valuable suggestion. It is our honor to improve the quality of the manuscript by adding and citing these research articles within lines 176–180 as references [53] and [54], which are highlighted in yellow.
“For example, Cheng et al. developed an injectable lubricative hydrogel based on a hydration-lubrication mechanism and dynamic disulfide bonds via a pseudo-click reaction for tendon repair [53]. Similarly, Ye et al. functionalized a linear DNA strand with a responsive luminescent active molecule through disulfide bonds for wound-healing applications [54].”
Comment 2:
Lines 565–569: The advantages of oxime click chemistry over thiol-based linkages need deeper explanation. Support with examples from tissue engineering and sensor-responsive hydrogels. Consider referencing, Electro spun green fluorescent-highly anisotropic conductive Janus-type nanoribbon hydrogel array film for multiple stimulus response sensors; doi: https://doi.org/10.1016/j.compositesb.2024.111933 & doi: 10.1038/s41467-024-51734-7.
Response 2:
We appreciate your valuable suggestion. It is our honor to improve the quality of the manuscript by adding and citing these research articles within lines 575–584 as references [140] and [141], which are highlighted in yellow.
“Based on recent reports, researchers have demonstrated the superiority of oxime click chemistry over thiol-based crosslinking strategies. Oxime click chemistry provides higher chemo selectivity, greater stability under physiological conditions, and excellent biorthogonality compared to thiol-based linkages, which are often prone to oxidation and require radical initiators. In tissue engineering, oxime-based hydrogels enable mild, initiator-free gelation with dynamic covalent bonds that impart self-healing and adaptability, thereby outperforming thiol–ene systems that rely on photo initiators. Recent advances in both sensor-responsive hydrogels and tissue engineering further highlight the versatility of oxime chemistry for multifunctional biomedical applications [140, 141].”
Comment 3:
Lines 923–927 – Strengthen the wound healing section by integrating recent findings on hydrogel–immune cell interactions and bioactive modulation of the healing cascade. Suggest citing,
Response 3:
We appreciate your kind response. The suggested citation was not included in this helpful comment. However, the authors have applied your suggestion in the main manuscript, and the changes are highlighted in yellow.
“Recent advancements in bioactive modulation demonstrated the role of hydrogels as bioactive materials in activating platelets and regulating the chemotaxis of immune cells, and controlling cell expression[194].”
Comment 4:
Lines 1162–1170: The bioink design discussion should include recent developments in multifunctional and bioinspired hydrogel bioprinting. Suggest citing, (a) doi: https://doi.org/10.1016/j.seppur.2024.130597(b)doi:https://doi.org/10.1016/j.cis.2025.103470
Response 4:
Many thanks. Authors applied this suggested citation in the main manuscript which are highlighted in yellow.
“Recently, diverse reaction techniques have facilitated the creation of new innovative multifunctional hydrogel biomaterials, identified as bioinks for 3D bioprinting [214]. The hydrogels provide a crucial milieu by replicating the extracellular matrices present in natural tissue structures. Diverse hydrogels, both natural and manufactured, demonstrate multifunctional attributes including biocompatibility, elevated mechanical strength, pH responsiveness, and thermal sensitivity. These hydrogels, when integrated with nanomaterials, composite materials, growth factors, and biomolecules, have been used to create complex 3D structures and functional tissue constructions using 3D bioprinting technology for tissue engineering applications [215].”
Comment 5:
Ensure consistent abbreviation usage (e.g., CuAAC, SPAAC).
Response 5:
Appreciate your consideration. All of the Cu-AAC and SP-AAC are replaced with CuAAC and SPAAC.
Comment 6:
Revise figure captions to be more descriptive, highlighting mechanisms or applications.
Response 6:
Special Thanks. All of the figure caption are revised as your kind comment. In the main manuscript, the changes are highlighted in yellow
“Figure 2. Illustration of functional groups and dynamic covalent chemistries employed to design injectable click chemistry hydrogels, where reagents of matching colors undergo specific reactions under defined conditions to form crosslinked networks with tunable properties, self-healing ability, and excellent biocompatibility for biomedical applications.”
“Figure 3. The multidisciplinary pillars of modern regenerative medicine: showcasing advanced research in skin wound healing, corneal and ocular repair, cartilage and bone regeneration, neural scaffold engineering for spinal cord injuries, and the transformative technology of 3D bioprinting and fabrication.”
Comment 7:
Improve readability by reducing redundancy and polishing sentence structure.
Response 7:
Special Thanks for this valuable comment. The main manuscript is reviewed, refined and improved to improve readability.
Reviewer 2 Report
Comments and Suggestions for Authors
The review paper title “Click Chemistry-Based Hydrogels for Tissue Engineering” is well written. However, there a few suggestions to make it better:
1) In the abstract use of abbreviations is not recommended.
2) Line 30—38 please add more references.
3) in introduction, please explain how the present review is different from the review already present on the topic.
4) Ref 223 is corrupted. Please add again.
5) can you add some figures for better representation
6) can you add a schematic showing reactions involved in various click reactions
Author Response
Comment 1:
In the abstract use of abbreviations is not recommended.
Response 1:
Appreciate your kind response. All of the abbreviations are removed from the abstract.
Comment 2:
Line 30—38 please add more references.
Response 2:
As your comment, authors cited some related references which are highlighted in yellow.
Comment 3:
In introduction, please explain how the present review is different from the review already present on the topic.
Response 3:
Appreciate your kind words. Authors change the last paragraph in the introduction to cover your helpful comment toward increasing the quality and novelty of this work. This change is highlighted in yellow.
“While the application of click chemistry for fabricating hydrogels has been extensively documented in previous reviews, with their focus largely on fundamental reactions, basic material properties, and established models, this review builds upon that foundation to offer a distinct and timely contribution. We move beyond these established concepts to critically explore the cutting-edge potential of click-crosslinked hydrogels, with a distinct focus on emerging applications that point toward the future of the field. This includes a dedicated exploration of their transformative role in precision medicine, spatiotemporally controlled drug delivery systems, and the bio fabrication of hierarchically complex tissues. To contextualize their versatility, we first introduce common click reactions and highlight their parallel advancements in fields like nucleic acid chemistry, a connection often overlooked in tissue engineering-focused articles. Furthermore, we synthesize these insights to provide a forward-looking roadmap, offering practical design considerations and clear future perspectives aimed at empowering researchers to develop the next generation of clinically impactful and translational hydrogel-based therapies.”
Comment 4:
Ref 223 is corrupted. Please add again.
Response 4:
Appreciate your consideration. This reference is related to Catalyst-Free Click Chemistry for Engineering Chondroitin Sulfate-Multiarmed PEG Hydrogels for Skin Tissue Engineering. J. Funct. Biomater. 2022, 13, 45. https://doi.org/10.3390/jfb13020045”. Authors refine this citation in the main manuscript. The change is highlighted in the manuscript.
Comment 5, Comment 6:
Can you add some figures for better representation. Can you add a schematic showing reactions involved in various click reactions
Response 5,6:
Thanks for this kind response. Authors illustrated Scheme 1 in section 2 as a valuable scheme relating to Click Chemistry reactions based on their different mechanisms.
Reviewer 3 Report
Comments and Suggestions for Authors
The topic of hydrogels crosslinked via click chemistry is highly relevant and timely, given the growing interest in precision medicine, regenerative medicine, and advanced biomaterials. The manuscript is generally well-written and provides a clear overview of hydrogel systems and their biomedical applications. The focus on click chemistry is of significant interest to the readership and has the potential to offer novel insights. Nevertheless, several aspects could be further clarified or expanded to enhance the manuscript’s scientific rigor and overall impact before acceptance.
-
The phrase “opportunities for innovative innovations remain limited” (line 46–47) is somewhat redundant and could be revised for clarity. This statement would also benefit from supporting references that critically discuss why conventional chemical crosslinking has plateaued and what challenges remain unresolved.
-
While the review emphasizes regenerative medicine and tissue engineering, the introduction also points to broader applications (e.g., agriculture, biotechnology). A brief discussion on how progress in these areas may inform biomedical hydrogel design could broaden the scope and originality of the work.
-
An important point for improvement is the inclusion of schematic figures illustrating representative click reactions. At present, the discussion is text-heavy, which may limit accessibility for readers less familiar with the chemistry.
-
Consistency in terminology (e.g., biocompatibility, bioactivity, bioorthogonality) should be ensured, as some terms are introduced but not consistently defined.
-
While the manuscript provides an overview of click reactions, it would benefit from a more systematic summary of how click chemistry influences hydrogel performance compared with conventional crosslinking methods.
Author Response
Comment 1:
The phrase “opportunities for innovative innovations remain limited” (line 46–47) is somewhat redundant and could be revised for clarity. This statement would also benefit from supporting references that critically discuss why conventional chemical crosslinking has plateaued and what challenges remain unresolved.
Response 1:
We thank the reviewer for this helpful comment. The redundant phrase has been revised for clarity, and we have added a new sentence highlighting the challenges of conventional chemical crosslinking methods, with appropriate references.
Comment 2:
While the review emphasizes regenerative medicine and tissue engineering, the introduction also points to broader applications (e.g., agriculture, biotechnology). A brief discussion on how progress in these areas may inform biomedical hydrogel design could broaden the scope and originality of the work.
Response 2:
We thank the reviewer for this valuable suggestion. We have revised the introduction to briefly highlight how advances in non-biomedical fields, such as agriculture and biotechnology, have informed biomedical hydrogel design. Specifically, we added a section discussing how stimuli-responsive and eco-friendly hydrogels developed for agricultural and environmental applications provide design principles such as tunable functionality, biodegradability, and environmental responsiveness that have been adapted for biomedical use through click chemistry. Relevant references have also been included to support this addition.
Comment 3:
An important point for improvement is the inclusion of schematic figures illustrating representative click reactions. At present, the discussion is text-heavy, which may limit accessibility for readers less familiar with the chemistry.
Response 3:
Thanks for this kind response. Authors illustrated Scheme 1 in section 2 as a valuable scheme relating to Click Chemistry reactions based on their different mechanisms.
Comment 4:
Consistency in terminology (e.g., biocompatibility, bioactivity, bioorthogonality) should be ensured, as some terms are introduced but not consistently defined.
Response 4:
We thank the reviewer for this important comment. We have now carefully reviewed the manuscript to ensure consistent definition and usage of key terminologies such as "biocompatibility," "bioactivity," and "bioorthogonality." Each term is now clearly defined upon its first appearance in the text, and we have ensured its consistent application throughout the manuscript to avoid any potential confusion.
Comment 5:
While the manuscript provides an overview of click reactions, it would benefit from a more systematic summary of how click chemistry influences hydrogel performance compared with conventional crosslinking methods.
Response 5:
We thank the reviewer for this insightful comment. In response, we have revised Section 2 to provide a more systematic summary comparing click chemistry with conventional crosslinking methods. This includes highlighting how click chemistry enhances hydrogel performance in terms of tunable mechanical properties, biocompatibility, degradation control, and biofunctionalization, as well as its advantages in precision and biorthogonal reactions under physiological conditions. Relevant references have been added to support these points.
Round 2
Reviewer 1 Report
Comments and Suggestions for Authors
The manuscript titled “Click Chemistry-Based Hydrogels for Tissue Engineering” is clearly written, and presents a meaningful contribution to the field. The authors have thoroughly addressed the reviewers’ earlier comments, and I recommend its acceptance in the current form.